# PrefixMemory-Tuning: Modernizing Prefix-Tuning by Decoupling the Prefix from Attention

**Haonan Wang**[1,*]  **Brian K. Chen**[1,2,*]  **Siquan Li**[3,*]  **Xinhe Liang**[1]
**Hwee Kuan Lee**[1,2]  **Kenji Kawaguchi**[1]  **Tianyang Hu**[3,†]
[1]National University of Singapore   [2]Bioinformatics Institute, A*STAR
[3]The Chinese University of Hong Kong, Shenzhen

## ABSTRACT

Parameter-Efficient Fine-Tuning (PEFT) methods have become crucial for rapidly adapting large language models (LLMs) to downstream tasks. Prefix-Tuning, an early and effective PEFT technique, demonstrated the ability to achieve performance comparable to full fine-tuning with significantly reduced computational and memory overhead. However, despite its earlier success, its effectiveness in training modern state-of-the-art LLMs has been very limited. In this work, we demonstrate empirically that Prefix-Tuning underperforms on LLMs because of an inherent tradeoff between the contribution of input prompt and parameterized prefix within the attention head. This motivates us to introduce **PrefixMemory-Tuning**, an architecture that generalizes the principles of Prefix-Tuning while addressing its shortcomings by shifting the prefix module out of the attention head itself and improving its expressiveness. Our experiments show that, across diverse benchmarks, PrefixMemory-Tuning consistently outperforms existing Prefix-Tuning methods. Notably, it achieves competitive performance with modern PEFTs on several general benchmarks, highlighting a potential extension of Prefix-Tuning approaches to become state-of-the-art. Our findings suggest that by overcoming its inherent limitations, Prefix-Tuning can remain a competitive and relevant research direction in the landscape of parameter-efficient LLM adaptation.

## 1 INTRODUCTION

Large Language Models (LLMs) have advanced at a remarkable pace in recent years, driven primarily by larger model architectures and bigger training datasets (Kaplan et al., 2020; Rae et al., 2022). This rapid progress, however, comes with soaring computational costs, making full-parameter fine-tuning on state-of-the-art models prohibitively expensive for all but the biggest players. To address this, parameter-efficient fine-tuning (PEFT) methods have been introduced. One such approach is Prefix-Tuning (PT) (Li & Liang, 2021), a technique which prepends trainable vectors to future inputs of each attention layer in the transformer. PT is both computationally cheap and effective, matching or even surpassing more complex methods in several early benchmarks.

However, as LLMs have scaled to unprecedented depths and parameter counts, PT has struggled to retain its effectiveness, leading to its replacement by newer methods such as LoRA (Hu et al., 2021) and GaLore (Zhao et al., 2024). The rapid pace of LLM research often leads to methods being abandoned before their limitations are fully understood or addressed. In the case of PT, its decline in popularity may reflect a lack of deeper investigation and adaptation. Despite having inherent advantages which are not directly reflected in performance, such as interpretability and relation to concepts such as memory, its poor performance has prevented further exploration of the methodology. This motivates a re-examination of PT in the context of modern LLMs—where its apparent limitations may be addressed and remedied.

---

*Equal contribution.
†Correspondence to Tianyang Hu. Email: `hutianyang@cuhk.edu.cn`.

Earlier studies have primarily attributed PT's underperformance to its inability to reshape attention patterns within attention heads (Petrov et al., 2023). We revisit this claim and show empirically that, while this explanation may hold for shallower transformers, it does not generalize to deeper architectures typical of modern LLMs. In this work, we argue that the real reason PT performs sub-optimally is its inherent tradeoff between prefix and input contribution. When the prefix is long relative to input length, the model risks losing input specificity and being dominated by the prefix. When the input is long relative to prefix length, the impact of the prefix is greatly diminished. This tradeoff is a result of prefixes being included in the softmax normalization term in the attention head.

Motivated by this, we build on previous work (Chen et al., 2024) to propose PrefixMemory-Tuning (PMT), which relocates the prefix outside the attention head by approximating it with an external module consisting of a trainable matrix $M$ and representation function $\phi(\cdot)$. Diagnostic experiments suggest that PMT is substantially more expressive than standard PT, serving as a proof of concept for our decision to shift the prefix outside the attention head. We also provide a unified overview of the design choices involved in externalizing the prefix, discussing how future work can build upon our preliminary approach when developing more advanced context-based methods.

Empirically, we evaluate PMT in both few-shot adaptation and application-scale settings across diverse preference alignment and math reasoning benchmarks. Against strong baselines ,e.g., LoRA and full fine-tuning, PMT is consistently competitive. Regular PT flounders in comparison. Our work presents the following key contributions.

- We demonstrate empirically that Prefix-Tuning performs poorly on modern LLMs because of an inherent tradeoff between input and prefix contribution within the attention head.
- We introduce PrefixMemory-Tuning, a novel architecture based on Prefix-Tuning that isolates the prefix module outside of the attention head. PrefixMemory-Tuning includes further modifications to improve the expressivity of the module. We provide a unified overview of our decision-making process in constructing PMT to guide users when constructing future context-based methods.
- We perform extensive experiments to show the efficacy of PMT. Our experiments show that, in the few-shot setting, PMT is competitive with state-of-the-art approaches such as LoRA—achieving an average absolute improvement of **8.1%** over LoRA and **29.4%** over Prefix-Tuning across all six evaluated settings.

This serves as a proof of concept that, when the prefix information is isolated from the attention head like in PMT, prefix-tuning methods can serve as a viable alternative to current modern methods and is an exciting future area of research.

## 2 RELATED WORK

**Weight-Based PEFT Methods.** LoRA (Hu et al., 2021) is one of the most widely adopted weight-based PEFT methods, injecting small trainable low-rank matrices into transformer layers while freezing the original weight matrices so that the effective updates lie in a low-dimensional subspace. QLoRA (Dettmers et al., 2024) further improves memory efficiency by applying low-rank adapters on top of a 4-bit quantized base model. LoRA+ (Hayou et al., 2024) does *not* change the low-rank parameterization itself; instead, it provides a theoretical analysis showing that using identical learning rates for the LoRA matrices $A$ and $B$ can hinder efficient feature learning in wide networks, and proposes differentially scaled learning rates with an optimally derived ratio for these adapter matrices. These methods primarily modify linear layers within transformer blocks and thus only implicitly affect the attention mechanism through weight updates.

**Context-Based PEFT Methods.** In contrast to weight-based methods, context-based PEFT methods directly alter the input context provided to LLMs while keeping the backbone weights frozen. Representative approaches include P-Tuning (Liu et al., 2021; 2024), Prompt Tuning (Lester et al., 2021), and Prefix-Tuning (Li & Liang, 2021), which prepend continuous vectors to the input embeddings or inject learnable prefixes into the key-value pairs of attention layers. Empirically, prefix-based methods have been shown to achieve competitive performance with full fine-tuning in low-data or few-shot settings on a variety of generation tasks (Li & Liang, 2021), while maintaining strong parameter efficiency. More recent work such as DePT (Shi & Lipani, 2023) and ADePT (Tang et al., 2025) further improves the flexibility of context-based PEFT by decomposing soft prompts into a short prefix plus additional low-rank or feed-forward components, and adaptively allocating capacity across them for different tokens. In the vision domain, E$^2$VPT (Han et al., 2023) proposes an effective

and efficient visual prompt tuning scheme that injects prompts *inside* attention heads, showing that modifying how prompts interact with attention can significantly improve efficiency and performance. However, several studies have also reported that the performance of Prefix-Tuning saturates or even degrades as the prefix length increases for large-scale models (Ouyang et al., 2023; Petrov et al., 2023), which limits its effectiveness in learning tasks that substantially differ from the pre-training distribution. Our work is motivated by these scalability issues and aims to improve how prefix-like signals interact with the underlying attention computation.

**Feed-Forward Layers as Memory.** A complementary line of work views the feed-forward networks (FFNs) in transformers as an implicit key-value memory that stores abstract knowledge in model parameters. Geva et al. (2021) show that FFN layers in Transformer language models can be formulated as a large key-value memory, where rows of the first linear layer act as "keys" that detect textual patterns and the corresponding rows of the second linear layer serve as values that induce output token distributions. Subsequent work further analyzes how FFN layers construct predictions by composing concept-level sub-updates in the vocabulary space (Geva et al., 2022), and how specific *knowledge neurons* inside FFNs express relational facts that can be localized and edited (Dai et al., 2022). More recently, Qiu et al. (2024) conduct an empirical study on updating the key versus value matrices in FFNs for knowledge editing and fine-tuning, providing additional evidence that FFNs operate as memories that store high-level knowledge.

Despite these limitations, context-based methods exhibit several unique advantages beyond raw performance. They offer greater interpretability due to the explicit and externalized nature of learned prompts (Lester et al., 2021; Le et al., 2025), enable lightweight test-time adaptation through prompt retrieval or modification (Zhou et al., 2022; Yi et al., 2025), and serve as a form of non-parametric memory for storing task-specific information (Kossen et al., 2024; Dai et al., 2022). These properties highlight the broader potential of context-based PEFT methods and motivate the need for improved designs. Yet, for vanilla Prefix-Tuning, its weaker performance in challenging settings makes it difficult to fully exploit these advantages in practice. This work aims to modernize Prefix-Tuning so it remains relevant in the current LLM regime.

## 3 PRELIMINARIES

Transformer models were introduced to address sequence-to-sequence tasks and primarily consist of attention layers, feed forward networks, and other task specific modules (Vaswani et al., 2017). In this paper, we assume an input sequence $X = [x_1, \ldots, x_n]$ with token embeddings $x_i \in \mathbb{R}^d$ for all $i \in [n]$ such that $X \in \mathbb{R}^{n \times d}$.

### 3.1 THE ATTENTION MECHANISM

Attention modules are a key component of transformers which accepts the entire sequence as an input. Typically, attention layers consist of multiple heads, each with a separate set of parameters. For notational simplicity we focus on single headed attention. A single attention head takes the form:

**Definition 1 (Single-headed Attention)** *Given input* $X \in \mathbb{R}^{N \times d}$ *and trainable matrices* $W_Q, W_K \in \mathbb{R}^{d \times d_K}, W_V \in \mathbb{R}^{d \times d_V}$. *A single attention head takes the form:*

$$o_i^\top = \frac{\sum_{j \leq i} \text{sim}(q_i, k_j) v_j^\top}{\sum_{j \leq i} \text{sim}(q_i, k_j)}. \tag{1}$$

*Based on Katharopoulos et al. (2020), where $o_i$ is the $i$-th output token whilst $q_i = x_i W_Q$, $k_i = x_i W_K$ and $v_i = x_i W_V$ and $\text{sim}(q, k) = \exp(\frac{qk^\top}{\sqrt{d_K}})$ is a similarity score.*

### 3.2 PREFIX-TUNING

**Definition 2 (Prefix-Tuning)** *Prefix-Tuning (PT) is a form of parameter-efficient fine-tuning that prepends a sequence of vectors to the inputs. Given prefix $[s_1, ..., s_p]$, where $s_i \in \mathbb{R}^d$ for all $i$, and input $X$, the new prompt becomes $X^{pt} = [s_1, ..., s_p, x_1, ..., x_n]$. The vectors $\{s_i\}_{i=1}^p$ are then trained based on traditional gradient based methods while the rest of the model weights are frozen.*

Referring to Equation 1, the inclusion of prefix $[s_1, ..., s_p]$ yields the following output:

$$o_i^{pt\,\top} = \frac{\sum_{j\leq i}\text{sim}(q_i, k_j)v_j^\top + \sum_{j\leq p}\text{sim}(q_i, W_K s_j)(W_V s_j)^\top}{\sum_{j\leq i}\text{sim}(q_i, k_j) + \sum_{j\leq p}\text{sim}(q_i, W_K s_j)}. \tag{2}$$

Compared with full parameter fine-tuning and even most other PEFTs, prefix-tuning offers an extremely light-weight training approach. Research shows that prefix-tuning excels in low-data or few-shot settings when guiding the model to leverage a mix of its pretrained tasks (Li & Liang, 2021; Petrov et al., 2023).

## 4 LIMITATIONS OF PREFIX-TUNING IN LLMS

In the previous section, we note that PT is particularly effective when leveraging pretrained tasks. With the continual increase in the size and capability of large language models (LLMs), supported by an ever-expanding pretraining corpus, one might anticipate a corresponding rise in the prominence and effectiveness of PT. However, contrary to expectations, the adoption of Prefix-Tuning has significantly declined in recent years, as evidenced by its sparse implementation on state-of-the-art models available in repositories such as Hugging Face. This diminished popularity is primarily due to PT's underwhelming performance with larger and more complex models, which manifests in reduced accuracy and instability. As depicted in Figure 1, Prefix-Tuning consistently underperforms compared to LoRA on three commonly used generative classification benchmarks, despite introducing a similar number of new parameters (see Section 6 for details). With LoRA and other PEFTs consistently outperforming Prefix-Tuning on established benchmarks, the overall relevance and applicability of Prefix-Tuning methods have been increasingly called into question.

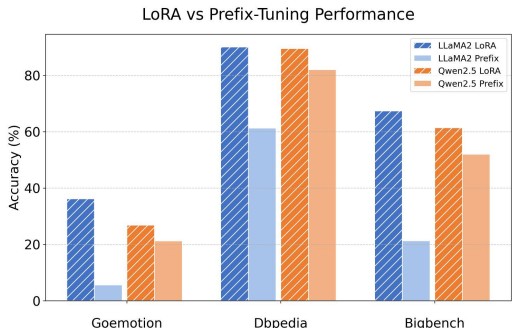

Figure 1: Performance comparison between Prefix-Tuning and LoRA.

### 4.1 DOES PREFIX-TUNING ALTER THE ATTENTION PATTERN?

So why doesn't Prefix-Tuning behave well on state-of-the-art LLMs? The popular stance is that PT cannot alter the attention distribution in the attention heads. As demonstrated in (Petrov et al., 2023), prefix-tuning is only capable of biasing the attention layer activations, which forms a severe limitation. This is shown to be true for single-layer transformers and shallow transformers in general. In this study, we argue that, while this analysis is indicative for shallow transformers, it does not capture how PT behaves on LLMs, which are deep multi-layer transformers. Our experiments in B.2 show that PT can modify the attention pattern of LLMs significantly, despite having bad performance. This leads us to believe that an inability to affect the attention pattern is not why PT performs badly.

### 4.2 TRADEOFF BETWEEN PREFIX AND INPUT SIGNIFICANCE

In this section, we argue that the fundamental limitation of Prefix-Tuning is the inherent tradeoff between the contribution of the prefix and the input. This can be observed by rewriting Equation 2 based on the work by (Petrov et al., 2023) as follows:

$$o_i^{pt\,\top} = (1 - \alpha_i)o_i^\top + \sum_{j\leq p}\alpha_{ij}v_j'^\top, \tag{3}$$

where $\alpha_{ij} = \frac{\text{sim}(q_i, W_K s_j)}{\sum_{j\leq i}\text{sim}(q_i, k_j) + \sum_{j\leq p}\text{sim}(q_i, W_K s_j)}$, $\alpha_i = \sum_{j\leq p}\alpha_{ij}$ and $v_j'^\top = W_V s_j'$.

Equation 3 shows that the output with Prefix-Tuning can be represented as a linear combination between the attention from the input $o_i$ and the attention from each prefix $v_j'$ with weights $\alpha_{ij}$. Prefix-Tuning mainly does two things: re-weights the original attention output and adds query-dependent bias vectors.

**When the prefix is long relative to input length:** In this case, we can expect the value of $\alpha$ to be large, which results in a greater change in the attention pattern since the base model's attention pattern is mainly dependent on $o_i$; this explains our observations in Figure 10. To further verify this, we conducted experiments with different prefix lengths and measured the changes in attention patterns using the REEF framework (Zhang et al., 2025). Our results in Table 10 confirm that as prefix length increases, the deviation from the base attention pattern grows. Details can be found in Appendix B.3. When $\alpha$ is large, the contribution from the input itself is smaller. The model then has reduced specificity regarding each input and risks being dominated by the prefixes. Too little significance may be placed upon the input itself.

This is further exacerbated by the fact that, as the length of the prefix increases, prefix-tuning is unable to make full use of the space spanned by the vectors $\{W_V s_i\}_{i=1}^p$. This phenomenon is also noticed by (Petrov et al., 2023) and is attributed to the competing optimization goals for the prefix $s_i$. The prefix both needs to grab attention through $W_K s_i$ and determine direction through $W_V s_i$.

**When the input is long relative to prefix length:** we can expect the value of $\alpha$ to be small. The opposite issue arises because when each $\alpha_i$ is small, the contribution of the prefix term is diminished. As LLMs get more and more capable, relying more on long sequences arising from techniques such as chain-of-thought reasoning (Wei et al., 2023), it is understandable for the effectiveness of prefix-tuning to be severely limited. Too little significance has been placed upon the prefix-tuning.

## 5    PREFIXMEMORY-TUNING: METHOD AND FRAMEWORK

### 5.1   MOTIVATION AND CONSTRUCTION

A key insight from Section 4.2 is that the trade-off between prefix and input contribution stems from the prefix's confinement within the softmax operator of the attention head. This motivates PrefixMemory-Tuning, a novel extension of PT, which represents a pilot attempt to bring the prefix information out of the attention head.

We first draw the terms containing the prefix information out of the attention head by splitting Equation 2 into:

$$o_i^{pt\ \top} = \lambda \frac{\sum_{j \leq i} \text{sim}(q_i, k_j) v_j^\top}{\sum_{j \leq i} \text{sim}(q_i, k_j)} + (1 - \lambda) \frac{\sum_{j \leq p} \text{sim}(q_i, W_K s_j)(W_V s_j)^\top}{\sum_{j \leq p} \text{sim}(q_i, W_K s_j)}, \qquad (4)$$

where $\lambda \in [0, 1]$ is a constant. This replaces the softmax regularization tradeoff, which is dependent on the length of the input and context, with a fixed convex linear combination similar to previous works (Munkhdalai et al., 2024; Wu et al., 2022). Then, we approximate the similarity metric $\text{sim}(\cdot, \cdot)$ with a kernel feature map $\phi$ such that $\text{sim}(\cdot, \cdot) \approx \phi(\cdot)^\top \phi(\cdot)$ and $\phi(\cdot) : \mathbb{R}^d \to \mathbb{R}^{d_\phi}$. We have

$$o_i^{pt\ \top} = \lambda \frac{\sum_{j \leq i} \text{sim}(q_i, k_j) v_j^\top}{\sum_{j \leq i} \text{sim}(q_i, k_j)} + (1 - \lambda) \frac{\phi(q_i)^\top \sum_{j \leq p} \phi(W_K s_j)(W_V s_j)^\top}{\phi(q_i)^\top \sum_{j \leq p} \phi(W_K s_j)}. \qquad (5)$$

A similar approach is used in Chen et al. (2024) to approximate in-context learning prompts, which has shown that the bias term $b_1 = \sum_{j \leq p} \phi(W_K s_j)(W_V s_j)^\top$ is capable of capturing contextual prompt or prefix information. The natural generalization of this step is to replace the bias $b_1$ by a more expressive, trainable matrix $M \in \mathbb{R}^{d_\phi \times d}$, and the analogous term $b_2 = \sum_{j \leq p} \phi(W_K s_j)$ by a trainable vector $N \in \mathbb{R}^{d_\phi}$, which yields:

$$o_i^{pt\ \top} = \lambda \frac{\sum_{j \leq i} \text{sim}(q_i, k_j) v_j^\top}{\sum_{j \leq i} \text{sim}(q_i, k_j)} + (1 - \lambda) \frac{\phi(q_i)^\top M}{\phi(q_i)^\top N}. \qquad (6)$$

In practice, due to the trainable nature of M and layer normalization, $\lambda$ can be absorbed into the trainable weights. Furthermore, $\phi(q_i)^\top N$ is no longer meaningful for regularization, so it can be removed. Therefore, the final attention output of the PrefixMemory-Tuning architecture found in figure 2 has the following form:

$$o_i^{PMT\ \top} = \frac{\sum_{j \leq i} \text{sim}(q_i, k_j) v_j^\top}{\sum_{j \leq i} \text{sim}(q_i, k_j)} + \phi(q_i)^\top M. \qquad (7)$$

## 5.2 CHOICE OF FEATURE MAP

Regarding the choice of $\phi$ there are many options which represent a tradeoff between expressivity and cost. The choice of $\phi$ is crucial for the performance of this method. A few from existing literature include $\phi(x) = \text{elu}(x)$ (Katharopoulos et al., 2020) and $\phi_W(x) = \text{ReLU}(Wx + b)$ (Mercat et al., 2024). In this study, as a proof of concept, we conduct experiments with $\phi(x) = \text{elu}(x)$ and $\phi(x) = \text{gelu}(x)$ due to ease of implementation. Experiments found in table 2 show that the choice of representation function, even between elu and gelu has a meaningful impact on performance. Other choices may offer more expressiveness and better performance, but would require significantly more detailed tuning and are left to future work.

**Remark 1 (Trainable feature maps)** *By choosing $\phi_W(x) = \text{ReLU}(Wx + b)$, the term $\phi_W(q_i)M$ becomes effectively a single-layer MLP. Depending on future choices for $\phi(\cdot)$, PrefixMemory-Tuning has the ability to be extremely expressive. As a universal approximator used alongside standard attention, this component intuitively has the potential to capture parameter updates and contextual information. Prior work (Chen et al., 2024) has shown that LoRA and SFT alone, when limited to parameter updates, is insufficient for capturing context information.*

*However, as PrefixMemory-Tuning is designed for parameter-efficient fine-tuning, our initial goal was to avoid introducing many learnable parameters. Furthermore, exploring issues like MLP initialization and training stability is a significant undertaking and is left for future research.*

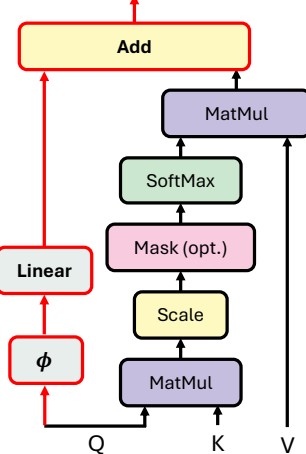

Figure 2: PrefixMemory in Scaled Dot-Product Attention.

## 5.3 A UNIFIED VIEW FOR CONTEXT BASED METHODS

This section outlines the design choices behind PT and PMT, offering the rationale for each to guide future implementation decisions. To arrive at PMT, there are two following decisions to be made:

1. Shift the prefix module out of the attention head
2. Approximate $\sum_{j \leq p} \text{sim}(., W_K s_j)$ by $\phi(\cdot)^\top M$

**Choice 1:** Shifting the prefix module out of the attention head is to avoid the limitations highlighted in Section 4.2. By doing so we avoid the $\alpha$ scaling on both the input and prefixes so there is no longer the same tradeoff between input contribution and prefix significance/contribution.

**Choice 2:** Replacing the original similarity metric by $\phi(\cdot)^\top M$ shifts the output from Equation 5 to Equation 6. By doing so, we lose some of the inherent structure of the attention mechanism. In return, we have an increase in model expressivity from the flexibility of a training matrix $M$. To verify this, since both PT and PMT can be viewed as adding query-dependent d-dimensional bias terms to the transformer, we calculate the covariate output matrices of the bias from each and find the respective eigenvalue decay. From Figure 3, we see that with PMT, the top eigenvalues corresponding to the main principal components are large and decay slowly compared to PT. This indicates that the output bias spans many principal components rather than collapsing onto a handful

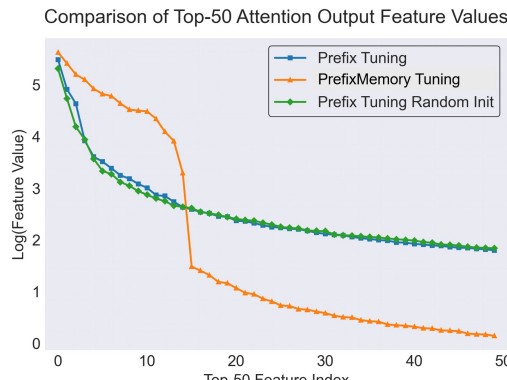

Figure 3: Spectrum of prefix representations.

of axes. In other words, PrefixMemory-Tuning adds a bias from a more diverse, high-dimensional subspace.

During the preparation of this paper, concurrent work (Meyer et al., 2025) was released, suggesting that prompt-tuning methods suffer from limited memorization capacity. In particular, there is an

Table 1: Fine-Tuning Method Performance Comparison (Accuracy %). Results across datasets and models; best-performing results are in boldface, highlighting the effectiveness of PrefixMemory-Tuning. LoRA+ results are added for comparison.

| Dataset | LLaMA2-7B-Chat | | | | | Qwen2.5-3B-Instruct | | | | |
|---|---|---|---|---|---|---|---|---|---|---|
| | PMT | Full | LoRA | LoRA+ | Prefix | PMT | Full | LoRA | LoRA+ | Prefix |
| GoEmotions | **45.2** | 32.7 | 36.2 | 39.4 | 5.6 | 37.3 | **37.8** | 26.8 | 32.7 | 21.2 |
| DBpedia | **92.7** | 92.6 | 90.1 | 91.9 | 61.3 | **96.9** | 94.4 | 89.5 | 94.4 | 82.0 |
| BigBench | **71.2** | 38.8 | 67.4 | 67.8 | 21.3 | **76.6** | 67.4 | 61.4 | 74.2 | 52.0 |

upper bound on the amount of new information that can be acquired through context prompts. This observation aligns with Choice 2 in our design, where we relax the previous architecture by replacing context prompts with $\phi(q)M$, thereby enhancing model expressivity and avoiding such constraints.

In future applications, users can choose to implement these choices in more sophisticated ways. For instance, our approach to shift the prefix module out of the attention head by replacing softmax regularization with linear convex combinations can be considered quite naive. We fully expect future work to build on this to achieve superior performance.

**Remark 2 (The Memory Perspective)** *We can view our method as explicitly treating the learnable matrix M as an internal memory store. Traditional context-based PEFTs, such as Prefix-Tuning, incorporate context memory by extending the KV inputs, tying the memory capacity to the prefix length. By linearizing attention and summing over the KV circuit, our approach decouples the memory capacity from sequence length and instead makes it proportional to the dimensionality of M, enabling more flexible storage of attention patterns. In practice, M allows the model to record and retrieve token interactions without altering the core attention weights by acting as an external memory module. This memory interface is both more direct and more parameter-efficient than auxiliary MLP-based memory modules, which typically require deep architectural changes, incurring higher costs.*

## 6 Experiments

To validate both the mechanism and the practical utility of PrefixMemory-Tuning. We structure our evaluation into two parts: **(i) diagnostic experiments** on well-instrumented datasets, enabling analyses of in-distribution (IID) accuracy and out-of-distribution (OOD) generalization that directly verify the previous analysis and mechanism advantages; and **(ii) large-scale post-training** that studies practical utility and efficiency at scale—treating PrefixMemory-Tuning as a PEFT component in two widely used application domains, *human preference alignment* and *math reasoning*. Across both parts, we show that PrefixMemory-Tuning outperform baselines while remaining training- and inference-efficient.

### 6.1 Mechanism Validation under Diagnostic Data

**Setup**. We evaluate on four generative classification benchmarks—BigBench (Suzgun et al., 2022), GoEmotions (Demszky et al., 2020), DBpedia (Kong et al., 2024), and Banking77 (Casanueva et al., 2020)—using two instruction-tuned models, LLaMA2-7B-Chat (MHA) (Touvron et al., 2023) and Qwen2.5-3B-Instruct (GQA) (Yang et al., 2024). We compare PrefixMemory-Tuning with full fine-tuning, LoRA (rank $r$=64) (Hu et al., 2021), Prefix-Tuning (virtual tokens $m$=32) (Li & Liang, 2021), and training-free in-context learning (Brown et al., 2020). For robustness, each configuration is run across five independent trials; in every trial we sample *one example per class* from the source dataset, fine-tune, and report in-distribution (IID) accuracy on the standard test split, averaged over the five trials. To assess out-of-distribution (OOD) generalization, models fine-tuned on BigBench/GoEmotions/DBpedia are evaluated on Banking77 without additional tuning. We leave full details in Appendix A.1.

**Overall adaptation performance.** Across three classification benchmarks, PrefixMemory-Tuning delivers superior or highly competitive accuracy while updating only a small fraction of parameters (Table 1). On BigBench, it reaches 71.2% with LLaMA2-7B-Chat and 76.6% with Qwen2.5-3B-Instruct, outperforming LoRA, Prefix-Tuning, and even full fine-tuning. On DBpedia, it attains top

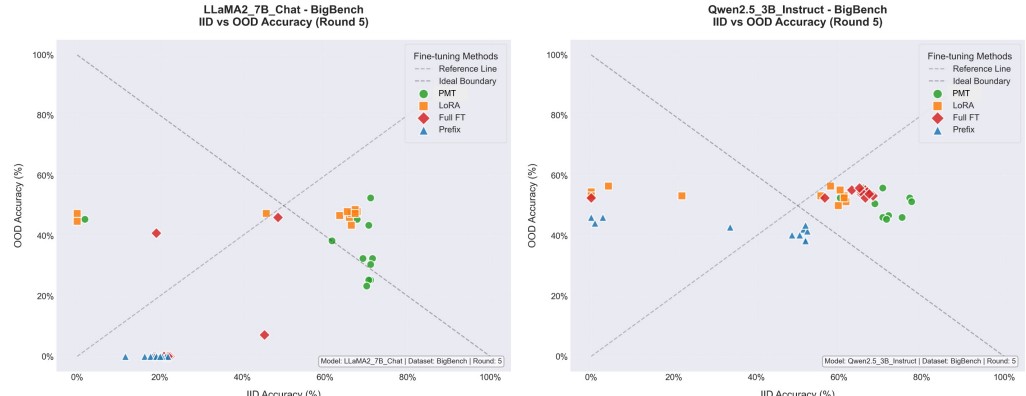

Figure 4: Pareto plots illustrating the trade-off between IID performance (on Bigbench) and OOD performance (on Banking77) for checkpoints of LLaMA2 and Qwen2.5 during training.

Table 2: Impact of feature-map activations on PrefixMemory-Tuning's accuracy (%).

| Dataset | PrefixMemory-Tuning (ELU) | | PrefixMemory-Tuning (GELU) | | PrefixMemory-Tuning (MLP Kernel) | |
|---|---|---|---|---|---|---|
| | LLaMA2-7B-Chat | Qwen2.5-3B-Instruct | LLaMA2-7B-Chat | Qwen2.5-3B-Instruct | LLaMA2-7B-Chat | Qwen2.5-3B-Instruct |
| GoEmotions | 45.2 | 37.3 | 47.0 | 38.7 | 43.6 | 35.7 |
| DBpedia | 92.7 | 96.9 | 93.2 | 96.4 | 95.0 | 95.0 |
| BigBench | 71.2 | 76.6 | 72.0 | 76.2 | 64.5 | 77.1 |

results (92.7% for LLaMA2, 96.9% for Qwen2.5). On GoEmotions, it remains robust (45.2% with LLaMA2-7B-Chat; 37.3% with Qwen2.5-3B-Instruct, within 0.5 points of full fine-tuning). Overall, PrefixMemory-Tuning consistently matches or exceeds strong baselines across models and tasks with far fewer updatable parameters. Details on parameter counts and performance across LoRA ranks are provided in the Appendix A.4.

**Balanced IID accuracy and OOD generalization.** Optimizing for in-distribution performance often degrades out-of-distribution robustness; PrefixMemory-Tuning avoids this trade-off. Using Banking77 as OOD and BigBench as IID, Pareto plots over checkpoints show PrefixMemory-Tuning consistently on (or near) the Pareto front for both LLaMA2-7B-Chat and Qwen2.5-3B-Instruct (Figure 4). This indicates that PrefixMemory-Tuning improves IID accuracy without sacrificing OOD resilience, yielding a better accuracy–robustness balance than alternative fine-tuning strategies. Additional datasets show the same trend (Appendix A.2).

**Stable across data sizes and attention types, with extra gains under GQA.** On BigBench with five incremental data rounds, PrefixMemory-Tuning maintains strong, smooth gains across scales and attention mechanisms (Figure 5). It matches or surpasses LoRA and full-parameter tuning for both LLaMA2-7B-Chat (standard attention) and Qwen2.5-3B-Instruct (grouped-query attention), with the largest improvements observed under GQA. These results suggest PrefixMemory-Tuning is architecture-friendly and data-scalable, making it practical for modern deployments where GQA is prevalent (Appendix A.3).

**Kernel Feature Map Design.** We study the kernel feature map $\phi(\cdot)$ in the prefix module by replacing ELU which is used in the previous setting with GELU. As shown in Table 2, GELU yields small but consistent gains on several tasks, suggesting $\phi$ matters. Heavier parameterizations (e.g., MLPs) are left to future work to preserve parameter efficiency.

## 6.2 PREFIXMEMORY-TUNING AT SCALE: PREFERENCE ALIGNMENT AND MATH REASONING

**Human Preference alignment.** We apply PrefixMemory-Tuning to align a 3B-parameter Qwen2.5 model with human preferences under different objectives, using 10k training samples for each. Specifically, we fine-tune the model with Supervised Fine-Tuning (SFT) on the Magpie-Ultra v0.1 instruction dataset (Xu et al., 2024), and with two preference optimization methods, Direct Preference Optimization (DPO) (Rafailov et al., 2023) and Simple Preference Optimization (SimPO) (Meng et al., 2024) on binarized UltraFeedback dataset (Cui et al., 2023). All experiments are run with the LLaMAFactory framework (Zheng et al., 2024) and evaluated using AlpacaEval 2, an auto-

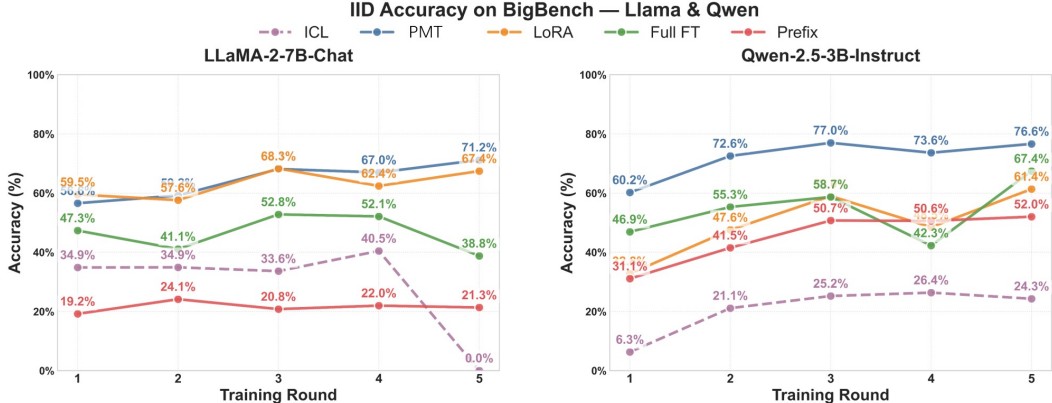

Figure 5: Performance over five incremental rounds of training data on BigBench. PrefixMemory-Tuning consistently matches or exceeds baselines, with the largest gains observed on Qwen-2.5-3B-Instruct.

Table 3: Accuracy (%) on open-ended math under CFT with different training sizes. For each training size and dataset, the higher of *LoRA* vs. PrefixMemory-Tuning is **bold**; red $(+\Delta)$ after PrefixMemory-Tuning shows the gain over LoRA at the same training size.

| Training samples | AMC23 | | Minerva-Math | | AIME24 | |
|---|---|---|---|---|---|---|
| | LoRA | PMT | LoRA | PMT | LoRA | PMT |
| 4 K | 40.0 | **42.5** (+2.5) | 16.5 | **20.2** (+3.7) | 10.0 | **13.3** (+3.3) |
| 10 K | 47.5 | **60.0** (+12.5) | 14.3 | **25.7** (+11.4) | 13.3 | **20.0** (+6.7) |
| 50 K | 47.5 | **60.0** (+12.5) | 23.9 | **62.5** (+38.6) | 16.7 | **23.3** (+6.6) |

matic win-rate benchmark for instruction-following models (Li et al., 2023). Across all objectives, PrefixMemory-Tuning delivers consistently higher win-rate improvements than LoRA as shown in Table 4. For example, under SFT the win-rate delta improves by **+0.76** with PrefixMemory-Tuning than +0.49 with LoRA; under DPO by **+4.66** vs +3.52; and under SimPO by **+1.74** vs +1.24. These results highlight the robustness and versatility of PrefixMemory-Tuning in preference alignment. Notably, the gains are strongest in the preference-based settings (DPO/SimPO), where PrefixMemory-Tuning notably outperforms LoRA. We also observe a small but consistent DPO is better than SimPO in our setup, likely due to SimPO's higher hyperparameter sensitivity (schrieffer-z, 2024).

**Math reasoning via CFT.** We adopt Critique Fine-Tuning (CFT) (Wang et al., 2025), training the model to critique noisy solutions rather than imitate gold answers. We fine-tune Qwen2.5-Math-7B on WebInstruct-CFT with 4K/10K/50K critique pairs and evaluate on AMC'23 (He et al., 2024), AIME'24, and Minerva-Math (Lewkowycz et al., 2022). Across all data scales, PrefixMemory-Tuning outperforms a strong LoRA baseline, with gains enlarging as data grows (Table 3); e.g., at 50K, Minerva-Math reaches 62.5% vs. 23.9% and AMC'23 60.0% vs. 47.5%. These results suggest PrefixMemory-Tuning scales reliably to advanced tasks.

## 6.3 PARAMETER EFFICIENCY AND COMPLEXITY ANALYSIS

We evaluate PrefixMemory-Tuning in terms of trainable parameters, memory footprint, training throughput, and inference latency. On BigBench, the peak fine-tuning memory usage is *comparable* (e.g., for LLaMA2-7B-Chat, LoRA with $r=32$ uses **16.5 GB** vs. **16.7 GB** for PrefixMemory-Tuning). The **training throughput**, measured on BigBench, results in Table 5 show that PrefixMemory-Tuning improves over LoRA on both the 3B and 7B models, and over Prefix-Tuning on the 7B model. The **inference latency** results on Qwen2.5-Math-7B and Qwen2.5-72B-Instruct, reported in Table 6, indicate that on the medium-scale 7B model, latency remains comparable to the Base and Prefix-Tuning variants, while on the large-scale 72B model, PrefixMemory-Tuning is slightly faster than Prefix-Tuning.

Table 4: AlpacaEval 2 win-rate deltas with 10K samples under SFT/DPO/SimPO. PrefixMemory-Tuning outperforms LoRA.

| Method | SFT | DPO | SimPO |
|---|---|---|---|
| LoRA | +0.49 | +3.52 | +1.24 |
| PrefixMemory-Tuning | **+0.76** | **+4.66** | **+1.74** |

Table 5: Training throughput. All numbers are iterations per second (*Iter./s*; higher is better).

| Model | PMT | LoRA ($r$=32) | Prefix-Tuning |
|---|---|---|---|
| LLaMA2-7B | **9.70** | 8.22 | 6.28 |
| Qwen2.5-3B | 7.57 | 6.54 | **7.94** |

Table 6: Inference latency on Qwen2.5-Math-7B and Qwen2.5-72B-Instruct. TTFT: time-to-first-token; TBT: time between tokens (lower is better); TPS: throughput (Token/s)

| Model | Method | Prefill (s) | Prefill TPS (tok/s) | Decode (s) | Decode TPS (tok/s) | TTFT (s) | TBT (s) |
|---|---|---|---|---|---|---|---|
| Qwen2.5-Math-7B | Base | 0.1769 | 13558 | 12.83 | 39.89 | 0.0259 | 0.02506 |
| Qwen2.5-Math-7B | PMT | 0.1803 | 13297 | 13.79 | 37.12 | 0.0279 | 0.02693 |
| Qwen2.5-Math-7B | PT | 0.1788 | 13409 | 13.16 | 38.89 | 0.0258 | 0.02571 |
| Qwen2.5-72B-Instruct | Base | 1.8378 | 1320 | 69.30 | 7.39 | 0.1314 | 0.1353 |
| Qwen2.5-72B-Instruct | PMT | 1.8396 | 1319 | 69.97 | 7.32 | 0.1327 | 0.1366 |
| Qwen2.5-72B-Instruct | PT | 1.8398 | 1319 | 70.89 | 7.22 | 0.1333 | 0.1384 |

## 7 CONCLUSION

In this work, we revisit Prefix-Tuning for modern LLMs and diagnose a core bottleneck: prefix signals are effectively trapped inside the attention head. We introduce PrefixMemory-Tuning, which decouples the prefix from attention head via a lightweight, query-conditioned module, preserving PEFT efficiency while restoring expressivity and stability. Across instruction following, preference alignment, and CFT-based math reasoning, PrefixMemory-Tuning consistently matches or surpasses strong adapters such as LoRA under small parameter budgets. This study positions context-methods as a viable path for future study and deployment. It is primarily a proof of concept to motivate further research into the inherent capabilities of context methods, such as how they tie into test-time scaling and training. We continue this discussion in Appendix D.

### ACKNOWLEDGMENTS

This material is based upon work supported by the Air Force Office of Scientific Research under award number FA2386-24-1-4011, and this research is partially supported by the Singapore Ministry of Education Academic Research Fund Tier 1 (Award No. T1 251RES2509).

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

# A    DETAILS ABOUT CONTROLLABLE EXPERIMENT

## A.1    EXPERIMENT SETUP

**Datasets.** We use four generative classification datasets:

- **(1) BigBench** (Srivastava et al., 2022; Suzgun et al., 2022): A comprehensive evaluation suite consisting of 23 challenging tasks. We focus on the *Date Understanding* task, formulated as a 6-class QA problem in which the model must choose one of six answer categories. For simplicity, we refer to this setting as BigBench.
- **(2) GoEmotions** (Demszky et al., 2020): A fine-grained emotion classification dataset containing 58K Reddit comments labeled with 27 emotion categories plus neutral (28 classes total). As the largest human-annotated English emotion dataset, GoEmotions covers a broad taxonomy of emotions. We cast this as a generative QA task: the model reads a comment and generates the corresponding emotion label.
- **(3) DBpedia** (Kong et al., 2024): A widely used ontology classification dataset consisting of Wikipedia abstracts assigned to 14 top-level classes. We formulate this as a generative QA task where the model must output the correct class name given an abstract.
- **(4) Banking77** (Casanueva et al., 2020): A challenging intent classification dataset designed for conversational systems, consisting of 13,083 customer service queries annotated across 77 categories. We formulate this as a generative QA task where the model must generate the correct label given a customer query.

**Training and Evaluation Protocol.** We assess each method's ability to quickly adapt to downstream tasks in a few-shot setting by fine-tuning on up to five independent rounds of minimal data. In each round, we randomly sample one example per class (6 examples for BIG-bench, 28 for GoEmotions, and 14 for DBpedia) to form the entire training set. After fine-tuning, we report in-distribution (IID) accuracy on each dataset's standard test split, averaging results over the five rounds to mitigate sampling variability. Since the ability to quickly adapt to new tasks often comes at the cost of generalization, we also evaluate out-of-distribution (OOD) performance using the Banking77 intent-classification dataset without additional fine-tuning. During inference, models receive a multiple-choice prompt listing all 77 Banking77 intents and must select the most appropriate label for each query. OOD accuracy is computed as the proportion of test queries correctly classified, measuring how effectively learned features generalize to unseen domains. We perform this evaluation independently for each of the five models fine-tuned on different source datasets.

**Models and Training Configuration.** We experiment with two pre-trained language models to assess architectural effects: LLaMA2-7B-Chat and Qwen2.5-3B-Instruct. The LLaMA2 series models employ the multi-head attention (MHA) (Vaswani et al., 2017) and Qwen2.5 use grouped-query attention (GQA) (Ainslie et al., 2023). GQA ties together query heads by sharing key/value projections, offering faster inference and lower memory usage, which allows us to examine if such architectural differences impact adaptation efficacy. Both models are used in their instruction version in order to test the OOD performance. We fine-tune these models using the AdamW (Loshchilov & Hutter, 2017) optimizer with a small learning rate and a fixed number of training steps (4000 steps). All methods use the same small batch size (batch size = 2).

**Baselines.** We compare PrefixMemory-Tuning against several baseline approaches for adapting large language models, covering both parameter-efficient and traditional full fine-tuning, as well as a training-free prompt-based baseline:

- **Full Fine-Tuning**: All model parameters are fine-tuned on the minimal training set for each round. This represents the conventional approach where all weights of models are updated.
- **Low-rank adaptation (LoRA (Hu et al., 2021))**: LoRA freezes original model parameters and introduces trainable low-rank update matrices into each Transformer layer. Only these small rank-$r$ matrices are learned, substantially reducing the number of trainable parameters. We set $r = 64$ to approximately match the parameter count introduced by PrefixMemory-Tuning.
- **Prefix-Tuning (PT (Li & Liang, 2021))**: Standard prefix-tuning keeps all model weights fixed, learning only a continuous prefix vector that is prepended to the input at each Transformer layer. We follow the original implementation and set the prefix length $m = 32$.

- **In-Context Learning (ICL (Brown et al., 2020))**: Unlike the previous methods, ICL involves no parameter updates. Instead, the training examples are directly provided as demonstrations in the context at inference.

## A.2 MORE RESULTS: IN-DISTRIBUTION ACCURACYS AND OUT-OF-DISTRIBUTION GENERALIZATION

In this appendix, we provide additional Pareto plots to complement the analysis presented in expriments section. Specifically, Figures 6 and 7 illustrate the trade-offs between in-distribution (IID) and out-of-distribution (OOD) performances for fine-tuned LLaMA2-7B-Chat and Qwen2.5-3B-Instruct models across two additional datasets: GoEmotions and DBPedia.

Each plot shows the IID accuracy ($x$-axis) evaluated directly on the respective dataset's held-out test set, and the OOD accuracy ($y$-axis) evaluated on the Banking77 dataset without further fine-tuning. Points within each plot represent model checkpoints captured at different training intervals, with colors indicating the respective fine-tuning methods used.

Consistent with our observations in the main text, the proposed method frequently occupies positions near the Pareto front. This indicates its effectiveness in maintaining a balanced performance between achieving high accuracy on IID tasks and exhibiting strong generalization to OOD scenarios.

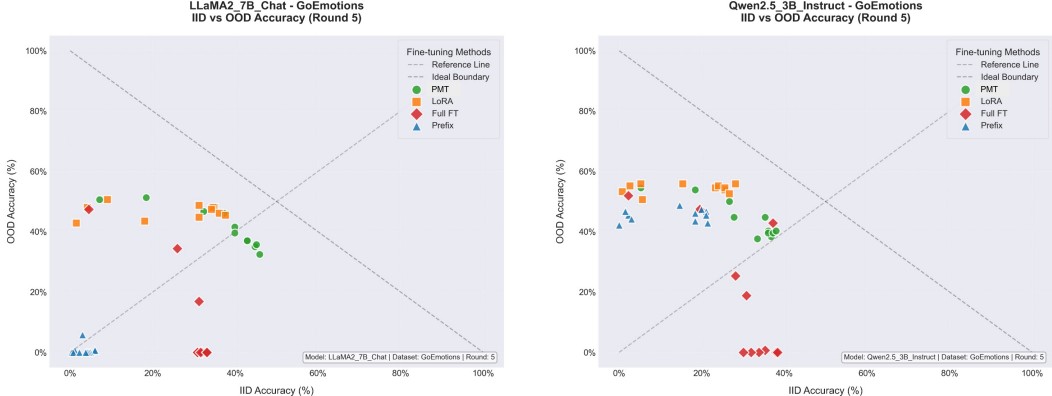

Figure 6: Pareto plots illustrating the trade-off between IID performance (on GoEmotions) and OOD performance (on Banking77) for checkpoints of LLaMA2-7B-Chat and Qwen2.5-3B-Instruct during training.

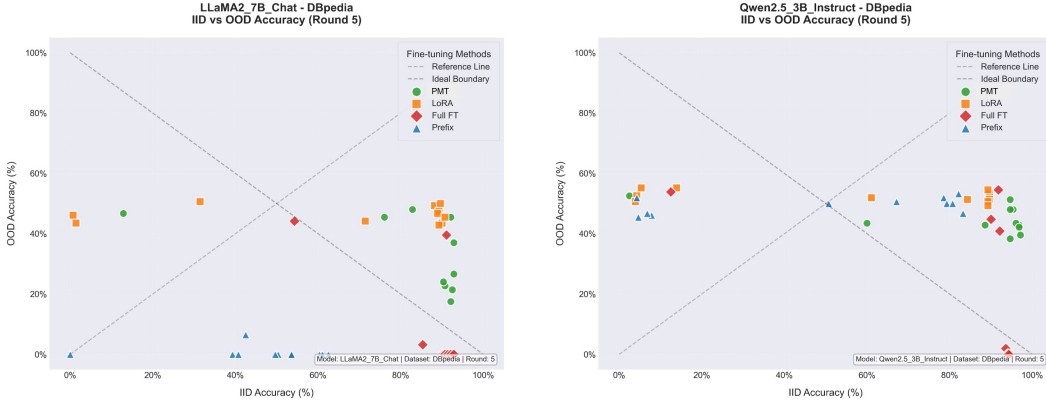

Figure 7: Pareto plots illustrating the trade-off between IID performance (on DBPedia) and OOD performance (on Banking77) for checkpoints of LLaMA2-7B-Chat and Qwen2.5-3B-Instruct during training.

### A.3 MORE RESULTS: PERFORMANCE ACROSS VARYING DATA SIZES AND ATTENTION MECHANISMS

To further validate the robustness and adaptability of PrefixMemory-Tuning across different tasks and attention mechanisms, we provide additional experiment results on two more datasets: GoEmotions and DBpedia. Similar to the main experiment, we incrementally increased the training set size across five rounds, fine-tuning two models—LLaMA-2-7B-Chat (multi-head attention, MHA) and Qwen-2.5-3B-Instruct (grouped-query attention, GQA)—using PrefixMemory-Tuning, Prefix-Tuning, LoRA, full-parameter fine-tuning, and the In-context Learning (ICL) baseline. Figure 8 and 9 illustrates the results across these additional datasets. Overall, these supplementary results reinforce our primary findings that PrefixMemory-Tuning scales effectively with data size and adapts particularly well to the grouped-query attention mechanism, outperforming existing parameter-efficient methods.

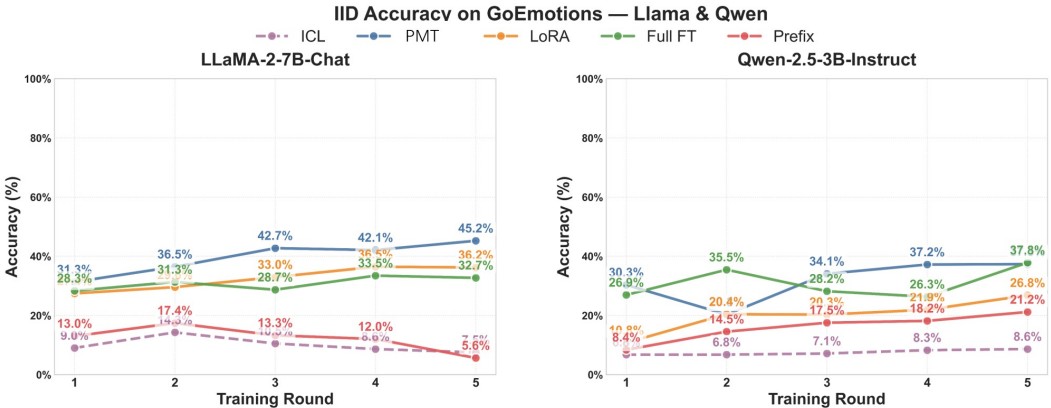

Figure 8: Performance comparison over five incremental rounds of training data on GoEmotions.

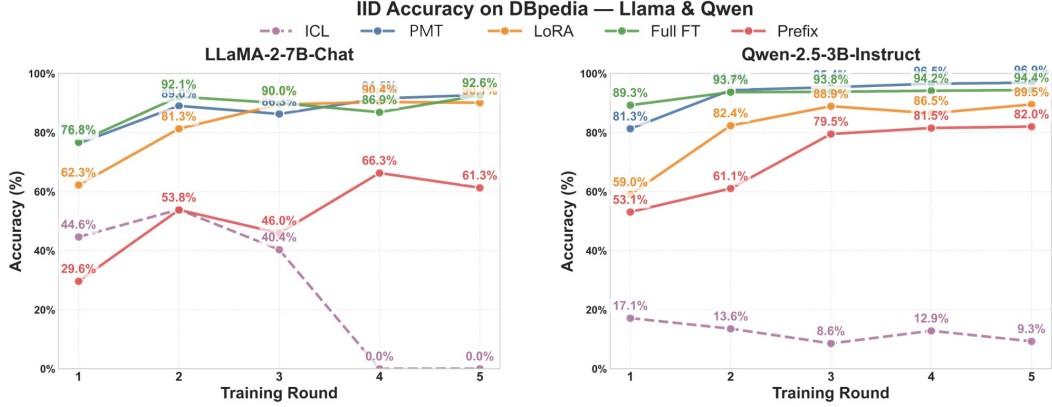

Figure 9: Performance comparison over five incremental rounds of training data on DBpedia dataset.

### A.4 MORE RESULTS: LORA RANK AND PARAMETER BUDGET

We sweep LoRA ranks $r \in \{32, 16, 8\}$ on GoEmotion, BigBench, and DBpedia under the same protocol. Lower ranks reduce parameters but hurt accuracy; higher ranks weaken parameter efficiency. With a budget comparable to LoRA $r \approx 32$, PrefixMemory-Tuning outperforms all ranks across datasets (Table 7).

Table 7: Accuracy (%) of LoRA at different ranks vs. PrefixMemory-Tuning.

| Dataset | Model | LoRA $r$=32 | LoRA $r$=16 | LoRA $r$=8 |
|---------|-------|-------------|-------------|------------|
| GoEmotions | LLaMA2-7B-Chat | 36.24 | 37.54 | 37.04 |
| | Qwen2.5-3B-Instruct | 26.85 | 27.85 | 26.65 |
| BigBench | LLaMA2-7B-Chat | 67.44 | 65.74 | 66.94 |
| | Qwen2.5-3B-Instruct | 61.45 | 60.85 | 62.05 |
| Dbpedia | LLaMA2-7B-Chat | 90.14 | 91.74 | 91.74 |
| | Qwen2.5-3B-Instruct | 89.55 | 88.05 | 92.05 |

Table 8: Trainable parameter counts and share of total for PrefixMemory-Tuning vs. LoRA.

| Model | Method | Trainable parameters (M) | Percentage (%) |
|-------|--------|--------------------------|----------------|
| LLaMA2-7B-Chat | LoRA $r = 64$ | 33.55 | 0.50 |
| | LoRA $r = 32$ | 16.78 | 0.25 |
| | LoRA $r = 16$ | 8.39 | 0.12 |
| | LoRA $r = 8$ | 4.19 | 0.06 |
| | PrefixMemory-Tuning | 16.91 | 0.25 |
| Qwen2.5-3B-Instruct | LoRA $r = 64$ | 14.75 | 0.48 |
| | LoRA $r = 32$ | 7.37 | 0.24 |
| | LoRA $r = 16$ | 3.69 | 0.12 |
| | LoRA $r = 8$ | 1.84 | 0.06 |
| | PrefixMemory-Tuning | 9.51 | 0.31 |

## B VERIFICATION EXPERIMENT SETUP

To better understand how different methods affect model behavior, we design three *comprehension-oriented experiments* that focus on analyzing attention patterns and internal representations. These experiments aim to shed light on the mechanisms and effects of each approach. For consistency and comparability, we use the GoEmotions dataset as the in-distribution (IID) dataset and the Banking77 dataset as the out-of-distribution (OOD) dataset across all experiments. The following subsections detail the setup of each experiment.

### B.1 SPECTRUM ANALYSIS OF PREFIX REPRESENTATIONS

In this experiment, we use Qwen2.5-3B-Instruct as the base model. We fine-tune two variants—prefix-tuning (with a prefix length of 32) and PrefixMemory-Tuning—on the GoEmotions dataset using identical training configurations and a consistent sampling strategy (5 rounds).

Let $F_b \in \mathbb{R}^{n \times d}$ denote the base model's final layer attention outputs for $n$ input tokens in total with representation dimension $d$, and $F_t \in \mathbb{R}^{n \times d}$ represent the corresponding fine-tuned model outputs. The representation effect (bias) matrix is computed as:

$$\Delta F = F_t - F_b$$

After normalization, we perform eigenvalue decomposition on the covariance matrix of representation effects:

$$\Sigma = \frac{1}{n-1} \Delta F^\top \Delta F = V \Lambda V^\top$$

where $\Lambda = \text{diag}(\lambda_1, ..., \lambda_d)$ contains eigenvalues ($\lambda_1 \geq ... \geq \lambda_d$), and $V$ is the orthogonal eigenvector matrix.

We concatenate examples from the GoEmotions test split into the input sequences and extract the self_attn.attn_output from the final layer. We then compute the corresponding attention outputs bias from the two fine-tuned variants, analyze their eigenvalue spectra, and visualize the top 50 eigenvalues to quantify how prefix tuning and our method alters the representation space geometry.

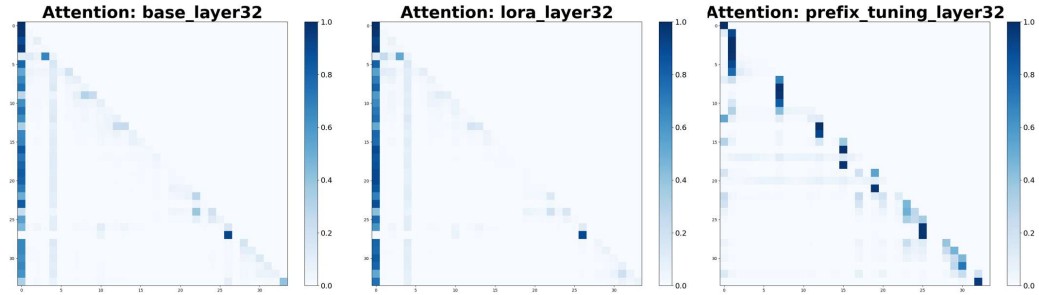

Figure 10: Attention Map of LLaMA2-7B-Chat, and its LoRA and Prefix-Tuning fine-tuned versions.

## B.2 ATTENTION PATTERN VISUALIZATION

This experiment examines how different fine-tuning methods impact attention behavior. We use LLaMA2-7B-Chat and Qwen2.5-3B-Instruct as base models, and fine-tune their respective prefix-tuning and PrefixMemory-Tuning variants using the same data and settings as in the previous experiment. We select one example each from the IID (GoEmotions) and OOD (Banking77) datasets as test inputs. For each model, we extract the `self.attn.attn_weight` from the final layer and visualize it as a heatmap to reveal attention patterns. For the prefix-tuning variants, we isolate the attention weights corresponding only to real tokens (excluding prefix tokens), normalize them, and then produce the heatmap visualization. The results of the experiment are found in figure 10. A systematic analysis of these attention patterns from the perspective of attention sinks (Xiao et al., 2023) would be an interesting direction for future work.

## B.3 REPRESENTATION SIMILARITY VIA CKA

Inspired by the REEF framework (Zhang et al., 2025), which utilizes centered kernel alignment (CKA) to quantify representation-level differences, we evaluate the similarity between base and fine-tuned models. The CKA similarity between two sets of representations $X$ (base model) and $Y$ (fine-tuned model) is computed as:

$$\text{CKA}(X, Y) = \frac{\text{HSIC}(X, Y)}{\sqrt{\text{HSIC}(X, X) \cdot \text{HSIC}(Y, Y)}},$$

where the Hilbert-Schmidt Independence Criterion (HSIC) is defined as:

$$\text{HSIC}(X, Y) = \frac{1}{(m-1)^2} \text{tr}(K_X H K_Y H).$$

Here, $H = I - \frac{1}{m} 11^T$ is the centering matrix, and $K_X$, $K_Y$ are Gram matrices with $(K_X)_{ij} = k(X_i, X_j)$ and $(K_Y)_{ij} = k(Y_i, Y_j)$, where $k$ is a kernel function (we use linear kernel in our experiments). $X_i$ denotes the $i$-th representation vector from layer outputs.

We use Qwen2.5-3B-Instruct as the base model, and obtain its prefix-tuning and PrefixMemory-Tuning variants using the same training data and setup. The TruthfulQA dataset is used for evaluation. Following the sampling and CKA computation protocol from the REEF paper, we extract decoder representations from the 18[th] layer of each model and compute the CKA similarity with the base model. This allows us to quantitatively assess how each method alters the internal representations while controlling for computational variance.

Table 9: CKA Similarity Between Different Methods And Base Model

| Method | CKA Similarity |
| --- | --- |
| Base Model | 1.0000 |
| LoRA | 0.9978 |
| PrefixMemory Tuning | 0.9432 |
| Prefix Tuning (32) | 0.8242 |

As shown in Table 9, we present the CKA similarity between the base model and the models fine-tuned using three PEFT methods: LoRA, PrefixMemory-Tuning, and prefix-tuning. It is evident that PrefixMemory-Tuning and LoRA exhibit notably different effects on the model's internal representations. Our proposed PrefixMemory-Tuning method induces more substantial shifts in the model's representation space, indicating a stronger impact on the model's expressive capacity. On the other hand, although prefix-tuning causes significant changes in the attention patterns, this also leads to much larger representation shifts, which may partly explain its relatively weaker downstream performance.

Table 10: CKA Similarity Between Prefix Tuning And Base Model

| Method | CKA Similarity |
|---|---|
| Base Model | 1.0000 |
| Prefix Tuning (16) | 0.8802 |
| Prefix Tuning (32) | 0.8242 |
| Prefix Tuning (64) | 0.7957 |

As shown in Table 10, we further examine how the prefix length affects the representation similarity between the prefix fine-tuned model and the base model under the same dataset and training settings. It is clear that as the prefix length increases from 16 to 64, the model's internal representations deviate more significantly from those of the base model, indicating that longer prefixes introduce more substantial changes in representation space.

In our experiments, since both Prefix-Tuning and PrefixMemory-Tuning only modify parameters within the self-attention mechanism—without affecting other components of the decoder layers—the resulting changes in representations can be regarded as a close approximation of changes in the attention pattern.

## C IMPLEMENTATION DETAILS

We implemented our experiments using PyTorch and trained our models utilizing the DeepSpeed optimization library with ZeRO Stage 3 to efficiently manage memory usage during training. To further optimize memory and computational efficiency, we offloaded both optimizer states and model parameters to CPU with pinned memory enabled, facilitating faster data transfers. Gradient communication and computation were overlapped, and contiguous gradients were enforced to enhance training throughput.

The AdamW optimizer was employed with a weight decay of 0.1, momentum terms set as $\beta_1 = 0.9$, $\beta_2 = 0.95$, and epsilon of $1 \times 10^{-8}$. Training was executed using automatic precision selection between FP16 and BF16 modes for optimal balance between performance and stability. The learning rate was held constant at $2 \times 10^{-5}$ throughout the training process. Each GPU processed a micro-batch size of one sample per step, while gradient accumulation was automatically managed to simulate larger batch sizes effectively. Gradient clipping was automatically controlled by DeepSpeed to maintain stable training dynamics.

For supervised fine-tuning (SFT) experiments, training was conducted using 2 GPUs, whereas human preference alignment experiments utilized 8 GPUs.

## D DISCUSSION

To conclude, in this work we argue that the reason why Prefix-Tuning has been ineffective when applied to modern large language models is that prefixes are "trapped" within the attention head. To remedy this, we introduce a novel architecture that generalizes upon existing Prefix-Tuning methods by approximating the prefix module and shifting it out of the attention head. Surprisingly, even with this slightly naive implementation, our model is able to match state-of-the-art methods such as LoRA on popular benchmarks in a few-shot setting, far outpacing previous prefix-tuning methods. We treat this as proof of concept that, if approached correctly, Prefix-Tuning methods can be competitive and are an exciting future avenue of research.

We also acknowledge the existing limitations of our work. Rather than presenting a clear alternative to existing PEFTs, PrefixMemory-Tuning is primarily a proof of concept. The design of our method has yet to be thoroughly ablated. For instance, this line of work can potentially be improved utilizing a more powerful choice of feature map $\phi$ such as a learnable one. Further studies are needed to test the limits of our method in more tasks and with more training objectives.

## E    LIMITATION

Despite the promising results demonstrated by PrefixMemory-Tuning, several areas remain open for exploration. Firstly, our implementation utilizes the kernel approximation for simulating attention, specifically the exponential linear unit (ELU). While this choice enabled efficient experimentation and a clear proof-of-concept demonstration, other feature mappings or kernel functions could potentially yield improved performance. Exploring more sophisticated kernel approximations or trainable kernel designs remains an exciting area for further enhancement of expressivity and effectiveness. Secondly, although PrefixMemory-Tuning effectively addresses the trade-off between prefix length and input specificity within attention heads, our experiments did not extensively explore the effects of varying internal dimensionalities or architectures of the externalized prefix module. Further studies investigating these architectural choices and their optimization could unlock additional performance gains. Thirdly, our evaluations were primarily conducted in supervised fine-tuning (SFT) and human alignment scenarios. Extending evaluations to contexts involving abundant data would provide deeper insights into PrefixMemory-Tuning's maximum capacity to acquire new knowledge. However, due to computational resource constraints at our institution, such comprehensive studies were beyond our current capabilities. We acknowledge this limitation and leave extensive evaluations to future research. Lastly, our evaluations are conducted on 3B–7B open-source models, which we treat as proxies for larger architectures. Extending the evaluation on larger models (e.g., more than 70B) would provide a more complete picture of PrefixMemory-Tuning. However, due to computational resource constraints in our academic environment, such comprehensive large-scale studies are beyond our current capabilities. We acknowledge this limitation and leave extensive large-scale evaluations to future work.

## F    BROADER IMPACTS

The introduction of PrefixMemory-Tuning offers significant positive impacts by making large language model (LLM) adaptation more efficient and accessible, thus enabling broader participation in AI research and application, particularly for resource-constrained communities and organizations. Additionally, by reducing computational requirements, PrefixMemory-Tuning contributes positively to sustainability efforts in AI development. On the other hand, the enhanced ease of adapting powerful LLMs also carries risks, such as potential misuse in generating misinformation or biased content. It is essential for researchers and practitioners to incorporate ethical practices, robust monitoring, and mitigation strategies to address these risks, ensuring that the societal benefits of PrefixMemory-Tuning significantly outweigh its potential negative impacts.

## G    LICENSES

We use standard licenses from the community. We include the following licenses for the codes, datasets and models we used in this paper.

Datasets & Benchmarks:

- BigBench (Srivastava et al., 2022): MIT
- GoEmotions (Demszky et al., 2020): Apache License 2.0
- DBPedia (Kong et al., 2024): Creative Commons 3.0
- Banking77 (Casanueva et al., 2020): MIT

Codes:

- LLaMA-Factory Zheng et al. (2024): Apache License 2.0

- Alpaca-eval Zheng et al. (2024): Apache License 2.0

Models:

- Qwen2.5-3B-Instruct (Yang et al., 2024): Apache License 2.0
- LLaMA2-7B-Chat (Touvron et al., 2023): LLaMA2 Community License

## H    LLM USAGE

We used large language models (ChatGPT and Gemini) as writing and formatting assistants. In particular, it helped refine grammar and phrasing, improve clarity, and suggest edits to figure/table captions and layout (e.g., column alignment, caption length, placement). The LLM did not contribute to research ideation, experimental design, implementation, data analysis, or technical content beyond surface-level edits. All outputs were reviewed and edited by the authors, who take full responsibility for the final text and visuals.

