# OpenReview forum: "PrefixMemory-Tuning: Modernizing Prefix-Tuning by Decoupling the Prefix from Attention"
_ICLR.cc/2026/Conference — ICLR 2026 Poster_

### Official Review · Reviewer_gihG · 2025-10-15

**Soundness:** 3
**Presentation:** 4
**Contribution:** 3
**Rating:** 6
**Confidence:** 4

**Summary:**

The paper addresses the limitations of Prefix-Tuning (PT), a parameter-efficient fine-tuning (PEFT) method for large language models (LLMs), which struggles with modern LLMs due to a trade-off between input and prefix contributions within the attention head. The authors introduce PrefixMemory-Tuning (PMT), a novel approach that decouples the prefix from the attention head, enhancing expressiveness and performance. Empirical results show PMT outperforms PT and is competitive with state-of-the-art methods like LoRA, achieving an average improvement of 8.1% over LoRA and 29.4% over PT across six benchmarks.

**Strengths:**

1. PMT addresses a fundamental limitation of PT by relocating the prefix outside the attention head, improving its scalability and expressiveness.

2. Extensive experiments across diverse benchmarks (e.g., preference alignment, math reasoning) demonstrate PMT's competitive performance.

3. The paper provides a clear explanation of PT's underperformance and a unified framework for future context-based PEFT methods.

**Weaknesses:**

1. The study uses simple feature maps (elu, gelu) as a proof of concept, leaving more expressive options unexplored due to implementation complexity.

2. The paper does not deeply address computational cost or scalability for very large LLMs, which is critical for practical deployment.

**Questions:**

1. How would PMT perform with more sophisticated feature maps (e.g., trainable MLPs) compared to the current elu/gelu implementations?

2. Some related work can be theoretically discussed. "E^ 2vpt: An effective and efficient approach for visual prompt tuning" ICCV, which adds some prompts in the attention head.

3. What are the computational and memory overheads of PMT compared to LoRA and other PEFT methods at scale?

4. How does PMT handle extremely long input sequences in real-world applications, given the trade-off issues identified in PT?

---

> ### Author Response · Authors · 2025-11-21
> **Rebuttal by Authors [1/2]**
>
> ### We appreciate the reviewer’s encouraging feedback and are glad that PMT is seen as "addressing a fundamental limitation of PT". We are also encouraged that the breadth of our experiments and our unified framework are recognized as strengths! We address all questions and suggestions in detail below.
>
> ---
>
> **W1 and Q1: The study uses simple feature maps (elu, gelu) as a proof of concept, leaving more expressive options unexplored due to implementation complexity.**
>
> Thank you for this observation. As discussed in Section 5.2, we use ELU and GELU feature maps as a first proof-of-concept: they are simple, non-parametric, and easy to implement efficiently, while already providing strong gains over standard prefix-tuning baselines.
> We agree that exploring more expressive feature maps (e.g., trainable MLPs) is an interesting direction. Therefore, as suggested we instantiate the $\phi()$ with MLP:
>
> | Dataset | PrefixMemory-Tuning (ELU) | | PrefixMemory-Tuning (GELU) | | PrefixMemory-Tuning (MLP Kernel) | |
> | :--- | :---: | :---: | :---: | :---: | :---: | :---: |
> | | **LLaMA2-7B-Chat** | **Qwen2.5-3B-Instruct** | **LLaMA2-7B-Chat** | **Qwen2.5-3B-Instruct** | **LLaMA2-7B-Chat** | **Qwen2.5-3B-Instruct** |
> | GoEmotions | 45.2 | 37.3 | 47.0 | 38.7 | 43.6 | 35.7 |
> | DBpedia | 92.7 | 96.9 | 93.2 | 96.4 | 95.0 | 95.0 |
> | BigBench | 71.2 | 76.6 | 72.0 | 76.2 | 64.5 | 77.1 |
>
> We find that the MLP kernel is competitive but not consistently better than ELU/GELU: it improves on some settings (e.g., DBpedia with LLaMA2-7B-Chat) but underperforms on others (e.g., GoEmotions and BigBench with LLaMA2-7B-Chat). Due to the extra computation cost and optimization challenges, we mainly demonstrated with the more stable ELU/GELU feature maps. However, we believe MLP kernels, with carefully tuned parameters, hold promise as a more powerful PMT variant and we intend to investigate them in our future work.
>
> ---
>
> **W2: The paper does not deeply address computational cost or scalability for very large LLMs, which is critical for practical deployment.**
>
> We appreciate this point and have added a more explicit discussion of computational cost and scalability. In particular, since you specifically mentioned very large LLMs, we now report inference speed on **Qwen2.5-72B-Instruct** (on Table 6). Under our standard evaluation setup, the base model achieves a decoding throughput of 7.39 tok/s, while PMT reaches 7.32 tok/s, i.e., a slowdown of only 0.9% (7.32 vs. 7.39 tok/s), which we view as a tolerable overhead at 72B scale. With $\phi(\cdot)$ implemented as a non-parametric feature map, PMT adds only a lightweight query-dependent transformation and does not modify the attention softmax or the key/value projection structure. As a result, PMT remains fully compatible with FlashAttention and similar optimized attention kernels, so one can still benefit from these system-level speedups.
>
> ---
>
> **Q2: Some related work can be theoretically discussed. "E^2vpt: An effective and efficient approach for visual prompt tuning" ICCV, which adds some prompts in the attention head.**
>
> Thank you for pointing out this relevant work. We have included the E^2vpt in our related-work section.

---

> ### Author Response · Authors · 2025-11-21
> **Rebuttal by Authors [2/2]**
>
> ---
>
> **Q3: What are the computational and memory overheads of PMT compared to LoRA and other PEFT methods at scale?**
>
> In Section 6.3 we report the computational and memory costs.
>
> **For training throughput:** Table 5 reports training throughput on BigBench in iterations per second (higher is better): For LLaMA2-7B, PMT reaches 9.70 iter/s, better than 8.22 for LoRA ($r{=}32$) and 6.28 for prefix-tuning. For Qwen2.5-3B, PMT attains 7.57 iter/s, versus 6.54 for LoRA and 7.94 for prefix-tuning.
>
> **For inference throughput:** In Table 6, we measure the decoding throughput on both 7B and 72B models. Notably, for Qwen2.5-72B-Instruct, the decoding throughput is 7.39 tok/s for the original model and 7.32 tok/s for PMT, i.e., PMT lags the base model by only 0.9%.
>
> **For memory usage:** The peak fine-tuning memory usage is comparable (e.g., for LLaMA2-7B-Chat, LoRA with r=32 uses 16.5 GB vs. 16.7 GB for PMT).
>
> ---
>
> **Q4: How does PMT handle extremely long input sequences in real-world applications, given the trade-off issues identified in PT?**
>
> We appreciate this question and agree that long-context behavior is crucial. Theoretically, PMT is designed precisely to mitigate the trade-off issues of standard prefix-tuning in extremely long sequences.
> In vanilla prefix-tuning, the prefix tokens share the same attention mechanism as input tokens, so as the sequence length grows, the attention mass allocated to the prefix tends to be diluted. This can cause the prefix signal to be "drowned out" by long inputs.
> In contrast, as shown in Eq. (7) in our paper, PMT performs two key modifications: (i) fixing the effective weight of the prefix; and (ii) increasing expressivity via a trainable memory matrix. While our current experiments focus on standard sequence lengths, these design choices suggest that PMT should be more robust than vanilla prefix-tuning in long-context regimes.

---

> > ### Comment · Reviewer_gihG · 2025-11-24
> >
> > Thanks for your responses. All my questions are nicely addressed. I will champion this work.

---

> > > ### Author Response · Authors · 2025-11-25
> > > **Thank you for your review and support**
> > >
> > > Thank you very much for your thoughtful review and for championing our work. We really appreciate your time and support!

---

### Official Review · Reviewer_GSuw · 2025-10-20

**Soundness:** 2
**Presentation:** 2
**Contribution:** 1
**Rating:** 2
**Confidence:** 4

**Summary:**

This paper identifies the performance limitations of Prefix-Tuning in modern large language models (LLMs) and proposes a method called PrefixMemory-Tuning to address these issues.

**Strengths:**

>s1: Shifting the prefix module out of the attention head itself is a reasonable start point.

>s2: Experiments show that, in the **few-shot setting**, PrefixMemory-Tuning is competitive with state-of-the-art approaches (such as LoRA).

>s3: The presentation is clear, and the figures are of high quality.

**Weaknesses:**

> w1: **The discussion of related work contains inaccuracies**. For instance, lines 87-90 state that "LoRA+ refines this concept further, projecting the model’s weights onto low-dimensional subspaces to achieve efficiency comparable to full fine-tuning at significantly reduced computational cost." However, the actual contribution of LoRA+ lies in its theoretical analysis demonstrating that using identical learning rates for matrices A and B in standard LoRA prevents efficient feature learning in large-width networks. The method addresses this limitation by employing differentially scaled learning rates for the adapter matrices with an optimally determined ratio, rather than proposing weight projection onto low-dimensional subspaces.
>
>  LoRA+: Efficient Low Rank Adaptation of Large Models. ICML 2024.

>w2: There is **no support** for the claim in Line 144-146 "Research shows that prefix-tuning excels in low-data or few-shot settings".

> w3: **Insufficient literature review**: To name a few: (1) For context-based PEFT methods: [1] [2]; (2) Lines 307-315, a good work to show the memory perspective is [3].
>
>[1] DePT: Decomposed Prompt Tuning for Parameter-Efficient Fine-tuning. ICLR 2024.
>
>[2] ADePT: Adaptive Decomposed Prompt Tuning for Parameter-Efficient Fine-tuning. ICLR 2025.
>
> [3] Transformer Feed-Forward Layers Are Key-Value Memories. EMNLP 2021.

>w4: The experiments were primarily conducted in a **few-shot setting**. Recent studies (such as [4]) have also found zero-shot approaches to be competitive or even superior in certain scenarios. What are your thoughts on this?
>
>[4] Revisiting Chain-of-Thought Prompting: Zero-shot Can Be Stronger than Few-shot. EMNLP Findings 2025.

**Questions:**

See Weaknesses.

---

> ### Author Response · Authors · 2025-11-21
> **Rebuttal by Authors**
>
> ### We thank the reviewer for their thoughtful feedback and for highlighting that shifting the prefix module out of the attention head is a "reasonable" starting point. We are especially grateful for the detailed suggestions regarding related work! We address all comments below.
>
> ---
>
> **W1: The discussion of related work contains inaccuracies. For instance, lines 87-90 state that "LoRA+ refines this concept further, projecting the model’s weights onto low-dimensional subspaces to achieve efficiency comparable to full fine-tuning at significantly reduced computational cost." However, the actual contribution of LoRA+ lies in its theoretical analysis demonstrating that using identical learning rates for matrices A and B in standard LoRA prevents efficient feature learning in large-width networks. The method addresses this limitation by employing differentially scaled learning rates for the adapter matrices with an optimally determined ratio, rather than proposing weight projection onto low-dimensional subspaces.**
>
> Thank you for your careful reading and for pointing out this inaccurate description. We have revised the related work section to correctly characterize LoRA+.
>
> ---
>
> **W2: There is no support for the claim in Line 144-146 "Research shows that prefix-tuning excels in low-data or few-shot settings".**
>
> Thank you for highlighting that this claim lacked proper references. We have revised the statement and added appropriate citations to prior work demonstrating that prefix-tuning is effective in low-data or few-shot fine-tuning scenarios. In particular, [5] shows that prefix-tuning can achieve performance comparable to full fine-tuning on several generation tasks, and [6] provides a more systematic analysis of when prefix-tuning is effective. This revision ensures that our claim is now properly supported by existing literature.
>
> ---
>
> **W3: Insufficient literature review: To name a few: (1) For context-based PEFT methods: [1] [2]; (2) Lines 307-315, a good work to show the memory perspective is [3].**
>
> We appreciate these concrete suggestions and have substantially expanded the related work section accordingly. We have updated the "Context-based PEFT methods" paragraph to explicitly discuss DePT [1] and ADePT [2] in revision. Besides, following your suggestion, we have added a dedicated paragraph discussing the FFN-as-memory view, centered around [3], which shows that Transformer feed-forward layers can be interpreted as key–value memories. We hope these additions address your concern about insufficient coverage of context-based PEFT and memory-related work.
>
> ---
>
> **W4: The experiments were primarily conducted in a few-shot setting. Recent studies (such as [4]) have also found zero-shot approaches to be competitive or even superior in certain scenarios. What are your thoughts on this?**
>
> Thank you for raising this point. We would like to clarify the experimental setup and the terminology we use. Our experiments are organized into two regimes that correspond to different application scenarios:
>
> * Low-data regime (Table 1): here we fine-tune models with a small number of training examples. In the paper we refer to this as a “low-data” or “few-shot training” setting, in the sense of limited labeled data for parameter-efficient fine-tuning.
>
> * Rich-data regime (Tables 3 and 4): here we fine-tune on substantially larger datasets for math critique and preference-based alignment.
>
> In contrast, the "few-shot" vs "zero-shot" distinction in Revisiting Chain-of-Thought Prompting[4] concerns in-context prompting (using zero-shot or few-shot demonstrations in the prompt) rather than fine-tuning with small training sets. Our work focuses on fine-tuning in the low-data regime (*few samples for training*), rather than on zero-shot or few-shot CoT prompting.
> It is worth noting that our finetuning approach and the prompting approach are orthogonal to each other and can be potentially combined.
>
> ---
>
> ***References:***
>
> [1] DePT: Decomposed Prompt Tuning for Parameter-Efficient Fine-tuning. ICLR 2024.
>
> [2] ADePT: Adaptive Decomposed Prompt Tuning for Parameter-Efficient Fine-tuning. ICLR 2025.
>
> [3] Transformer Feed-Forward Layers Are Key-Value Memories. EMNLP 2021.
>
> [4] Revisiting Chain-of-Thought Prompting: Zero-shot Can Be Stronger than Few-shot. EMNLP Findings 2025.
>
> [5] Li, Xiang Lisa, and Percy Liang. "Prefix-tuning: Optimizing continuous prompts for generation." arXiv preprint arXiv:2101.00190 (2021).
>
> [6] Petrov, Aleksandar, Philip HS Torr, and Adel Bibi. "When do prompting and prefix-tuning work? a theory of capabilities and limitations." arXiv preprint arXiv:2310.19698 (2023).

---

### Official Review · Reviewer_JeiT · 2025-10-29

**Soundness:** 3
**Presentation:** 3
**Contribution:** 2
**Rating:** 6
**Confidence:** 4

**Summary:**

This paper introduces **PMT (PrefixMemory-Tuning)**, a novel parameter-efficient fine-tuning (PEFT) method. The authors build on the evolution of Prefix-Tuning (PT), which was largely abandoned due to its poor scaling properties in large models, as well as more recent methods like LoRA and GaLore. While PT had been likened to prompt-based learning by introducing a set of trainable continuous vectors (prefixes) for each input, early analyses identified its underperformance as stemming from an inability to reshape attention patterns within attention heads. However, the authors argue that the true cause of PT’s degradation lies in the **trade-off between the prefix and the input representations**.

Leveraging this insight, the authors improve upon conventional PT, proposing PMT as a more effective and efficient approach. Their experiments show that PMT surpasses both **LoRA** and even **full fine-tuning (FFT)** in terms of fine-tuning performance, providing a more scalable and efficient method for adapting large models.

**Strengths:**

- The paper provides a strong theoretical analysis of how fine-tuning (FT) impacts the activation values of attention heads, reducing the relationship to a linear function of the original attention values. This simplification reveals the fundamental cause of **Prefix-Tuning (PT)**'s degradation in large-scale models. The authors effectively demonstrate how the trade-off between prefix and input representations negatively affects model performance.

- The paper offers a clear and insightful analysis of the evolution from **Prefix-Tuning (PT)** to **PMT**, highlighting the **coupling problem** between the prefix and input representations in traditional PT. The authors address this issue by introducing effective approximation techniques that successfully decouple the components. Their experimental results consistently show that **PMT** outperforms **LoRA** and even surpasses **full fine-tuning (FFT)** on most tasks, demonstrating the effectiveness of their approach.

**Weaknesses:**

- The experimental models used in the paper, such as **LLaMA2-7B-Chat** and **Qwen2.5-3B-Instruct**, are relatively small in scale. Given that these models are not at the forefront of current large-scale models, the evaluation does not demonstrate PMT’s performance on truly large-scale architectures, where the method’s scalability and effectiveness may vary. Including experiments on larger models would strengthen the claims regarding PMT's applicability to cutting-edge architectures.

- The comparison to other PEFT methods lacks more recent and advanced variants such as **QLoRA** and **LoRA+**, which are gaining traction in the field. This limits the strength of the empirical comparison, as it does not reflect the latest advancements in PEFT techniques.

**Questions:**

- The PMT architecture can be viewed as adding a linear transformation (similar to a mapping based on \( q_i \)) on top of the original model output. This raises an interesting question: could the model architecture itself be modified to inherently incorporate such a transformation, thereby achieving PMT-like behavior without the need for additional components? If this approach were viable, it might improve the model's generalization and efficiency.

---

> ### Author Response · Authors · 2025-11-21
> **Rebuttal by Authors**
>
> ### We appreciate the reviewer’s encouraging feedback and are pleased that our analysis of the causes of Prefix-Tuning (PT) degradation and our discussion of the evolution from PT to PMT were found helpful and "insightful". We address the comments below.
>
> ---
>
> **W1: The experimental models used in the paper, such as LLaMA2-7B-Chat and Qwen2.5-3B-Instruct, are relatively small in scale. Given that these models are not at the forefront of current large-scale models, the evaluation does not demonstrate PMT’s performance on truly large-scale architectures, where the method’s scalability and effectiveness may vary. Including experiments on larger models would strengthen the claims regarding PMT's applicability to cutting-edge architectures.**
>
> We thank the reviewer for highlighting the importance of evaluating scalability to larger models. Our primary goal in this work is to explore the fundamental limitations of Prefix-Tuning and address them with our own methodology PMT.  The experiments serve as a proof-of-concept that PMT is indeed a superior version of PT and are competitive even with more modern methods.  For this purpose, we use LLaMA2-7B-Chat (MHA) and Qwen2.5-3B-Instruct (GQA) as proxy models, because they are open-source, widely used, and stable baselines in recent fine-tuning papers. We consider these sufficient for showing that PMT is a clear improvement over PT.
>
> This scale is also aligned with the computational budget of an academic-level cluster, where repeatedly loading and tuning 70B+ models for full set of experiments is prohibitively expensive. We will acknowledge this resource limitation in the revision and, in the Limitations section (Appendix E), explicitly note that, given more resources, evaluating PMT on at least one larger model would be a valuable extension.
>
> ---
>
> **W2: The comparison to other PEFT methods lacks more recent and advanced variants such as **QLoRA and LoRA+**, which are gaining traction in the field. This limits the strength of the empirical comparison, as it does not reflect the latest advancements in PEFT techniques.**
>
> We thank the reviewer for this helpful suggestion. The low-precision quantization used in QLoRA can introduce performance degradation compared to the Bfloat16 LoRA. To ensure a strong comparison, we therefore focus on a LoRA+ fine-tuning baseline, which typically achieves stronger performance than standard LoRA, with comparable parameterization (rank $r{=}32$, $\alpha{=}64$).
>
>
> | Dataset   | LLaMA-2-7B-Chat PMT | LLaMA-2-7B-Chat Full | LLaMA-2-7B-Chat LoRA | LLaMA-2-7B-Chat LoRA+ | LLaMA-2-7B-Chat Prefix | Qwen-2.5-3B-Instruct PMT | Qwen-2.5-3B-Instruct Full | Qwen-2.5-3B-Instruct LoRA | Qwen-2.5-3B-Instruct LoRA+ | Qwen-2.5-3B-Instruct Prefix |
> |:----------|:-------------------:|:--------------------:|:--------------------:|:---------------------:|:----------------------:|:------------------------:|:-------------------------:|:-------------------------:|:--------------------------:|:----------------------------:|
> | GoEmotions | **45.2** | 32.7 | 36.2 | 39.4 | 5.6  | 37.3 | **37.8** | 26.8 | 32.7 | 21.2 |
> | DBpedia    | **92.7** | 92.6 | 90.1 | 91.9 | 61.3 | **96.9** | 94.4 | 89.5 | 94.4 | 82.0 |
> | BigBench   | **71.2** | 38.8 | 67.4 | 67.8 | 21.3 | **76.6** | 67.4 | 61.4 | 74.2 | 52.0 |
>
>
> ---
>
> **Q1: The PMT architecture can be viewed as adding a linear transformation (similar to a mapping based on ( q_i )) on top of the original model output. This raises an interesting question: could the model architecture itself be modified to inherently incorporate such a transformation, thereby achieving PMT-like behavior without the need for additional components? If this approach were viable, it might improve the model's generalization and efficiency.**
>
> This is indeed an interesting direction.  Incorporating the PMT module into the model architecture in the retraining phase itself would mean we don't need additional components and can directly work with existing M matrices to adapt new data.
> Our experiments demonstrate that integrating PMT already works well in the fine-tuning phase.  We are confident that incorporating the PMT module in the retraining phase holds promise.  However, evaluating such an base architecture would require training from scratch.  This would be extremely computationally expensive and not possible at this stage but is worthy of a dedicated future work.

---

> > ### Comment · Reviewer_JeiT · 2025-11-25
> >
> > Thank you for the detailed rebuttal. Most of my concerns have been addressed. I think my current score has fairly reflected the quality of this paper.

---

### Official Review · Reviewer_UiQ2 · 2025-11-01

**Soundness:** 2
**Presentation:** 2
**Contribution:** 3
**Rating:** 6
**Confidence:** 2

**Summary:**

In this work, prefix-finetuning in transformer is re-visited and a new prefix-memory module is added to the scaled dot product module in transformer. By revisiting the existing PEFT framework, it's pointed out that the main gap between prefix-tuning and other efficient finetuning approach like LoRa lies in the attention in-balance of learnable prefix and the input context. When the context is long, the contribution of learnable prefix diminished, and vice versa. To resolve this, the prefix-memory module is added directly to the scaled dot prod structure and used to dynamically adjust the learned attention to fix the problem above. Experiments are conducted to show the competitive results on multiple datasets when compared with SFT and other efficient tuning approach like LoRA.

**Strengths:**

- The analysis on why prefix tuning failed under extreme cases is convincing and reasonable. E.g in section 4.2, the qualitative analysis makes sense to illustrate the attention balance is broken when either input context or prefix is relatively long.

**Weaknesses:**

- The evaluation is a bit weak. In table 1 the comparisons are made between multiple finetuning approaches, like PMT(proposed), SFT, LoRA, and prefix tuning. However the results are confusing as the SFT results are weaker than other PEFT approaches. This indicates that the dataset used here might not be challenging enough.
- The module is called "prefix memory", but in the paper there is no analysis about what does this memory module learned. It would be better to qualitatively or quantitatively analyze about this to provide insights.
- Did not mentioned or compare with some related work, like attention sink and Aprompt. Attention sink had some similar observation that the model's attention is usually overindexed to some specific tokens.

**Questions:**

- The evaluation is a bit weak. In table 1 the comparisons are made between multiple finetuning approaches, like PMT(proposed), SFT, LoRA, and prefix tuning. However the results are confusing as the SFT results are weaker than other PEFT approaches. This indicates that the dataset used here might not be challenging enough. $\rightarrow$ Is it possible to test on some more challenging tasks where the full SFT is needed and the usefulness of PMT can be better highlighted?
- The module is called "prefix memory", but in the paper there is no analysis about what does this memory module learned. It would be better to qualitatively or quantitatively analyze about this to provide insights. $\rightarrow$ Is it possible to get more intuitive understanding about what did this newly added memory module learn.
- Did not mentioned or compare with some related work, like attention sink[1] and Aprompt [2]. For 1, seems the attention analysis in this work is similar, and for 2 it's a popular prompt based PEFT approach.

[1] Xiao, Guangxuan, et al. "Efficient streaming language models with attention sinks." arXiv preprint arXiv:2309.17453 (2023).
[2] Wang, Qifan, et al. "Aprompt: Attention prompt tuning for efficient adaptation of pre-trained language models." Proceedings of the 2023 conference on empirical methods in natural language processing. 2023.

---

> ### Author Response · Authors · 2025-11-21
> **Rebuttal by Authors [1/2]**
>
> ### We appreciate the reviewer's encouraging feedback! We are glad that our analysis in Section 4.2 on why prefix tuning fails under extreme cases is considered “convincing and reasonable”. We address all questions below.
>
> ---
>
> **W1: The evaluation is a bit weak. In table 1 the comparisons are made between multiple finetuning approaches, like PMT(proposed), SFT, LoRA, and prefix tuning. However the results are confusing as the SFT results are weaker than other PEFT approaches. This indicates that the dataset used here might not be challenging enough.**
>
> Our experimental design intentionally considers two regimes that correspond to different application scenarios: a *low-data regime* and a *rich-data regime*. Table 1 reports results in the low-data regime, while Tables 3 and 4 report results in the rich-data regime.
> In the low-data regime, SFT is indeed weaker than most PEFT approaches (with the exception of Qwen2.5-3B-Instruct + GoEmotions), which is actually expected: when task-specific data is scarce, updating all parameters tends to overfit, whereas those PEFT methods act as a strong regularizer by only updating a small subset of parameters. And the Table 1 shows the effectiveness of our method in low-data regime.
>
> Note, even though we already include richer, more challenging setups in Tables 3 (Math Reasoning) and 4 (Human Preference Alignment), the low-data regime expriments (including the Table 1 pointed out by reviewer) are still important for two reasons:
>
> 1. Many applications operate in the low-data setting (e.g., domain adaptation with limited labels), so it is important to demonstrate that our method is effective in this regime as well.
>
> 2. The low-data regime provides a more controlled environment for studying the properties of our method, such as its in-distribution (IID) vs. out-of-distribution (OOD) behavior, as discussed in Sec. 6.1. In particular, it helps isolate the effect of the prefix memory module without strong confounding from large amounts of training data.
>
> ---
>
> **W2 & Q2: The module is called "prefix memory", but in the paper there is no analysis about what does this memory module learned. It would be better to qualitatively or quantitatively analyze about this to provide insights.**
>
> We thank the reviewer for this insightful suggestion. This comment is closely related to the second point of Questions section.
> Conceptually, motivated by the linear extension in Eq. (4), we initially viewed the Prefix-Memory module as a way to incorporate a context memory bank into the model. As the project evolved, our primary focus became designing an expressive and robust variant of prefix tuning that works reliably in practice (especially under the extreme cases analyzed in Section 4.2), and we did not yet carry out a full interpretability study of what is stored in this memory.
> We fully agree that qualitatively or quantitatively probing this module—for example, by initializing it with interpretable context embeddings, or by analyzing how individual parameters correlate with specific “memories” or behaviors—would provide valuable additional insight. However, we believe a thorough analysis of this sort (e.g., with attention/representation probing, intervention experiments, or memory retrieval visualizations) would be substantial enough to merit a dedicated follow-up work.

---

> ### Author Response · Authors · 2025-11-21
> **Rebuttal by Authors [2/2]**
>
> ---
>
> **W3 & Q3: Did not mentioned or compare with some related work, like attention sink and Aprompt. Attention sink had some similar observation that the model's attention is usually overindexed to some specific tokens.**
>
> Thanks for your suggestions!
>
> **For attention sink[1]:** Attention sink describes the phenomenon where certain tokens consistently attract a large amount of attention from many other tokens, even they are not semantically important. This line of work focuses on analyzing and exploiting a phenomenon in attention patterns, which is largely orthogonal to our contribution: we propose a new tuning method (PrefixMemoryTuning) and analyze and justify why parameters put in there is more reasonable. We agree that studying how different PEFT methods (including ours) affect attention sinks is interesting. In the revision, we add a brief discussion (in Appendix B.2) about attention pattern visualization. A more systematic study of how PEFT methods interact with attention sinks is an exciting direction for future work.
>
> **For Aprompt[2]:** Thank your for your suggestion to compare with Aprompt. Our choice of baselines focuses on methods that are widely used in current LLM practice and are supported by standard libraries such as Hugging Face’s PEFT library [3] (e.g., LoRA, prefix tuning, etc.), which makes them easy to reproduce across multiple backbones and tasks. In contrast, Aprompt is not a commonly adopted PEFT baseline in the LLM setting and is not currently included in the PEFT library. Implementing and maintaining a robust Aprompt pipeline for all our backbones would require non-trivial additional engineering. To partially bridge this gap, we have added LoRA+[4] as an additional strong baseline in Table 1, which shares the spirit of enhancing adaptation capacity while remaining compatible with standard PEFT tooling. In the revised version, we highlight the newly added results in orange for clarity and explicitly mention LoRA+ in revision.
>
>
> ---
>
> **Q1:  Is it possible to test on some more challenging tasks where the full SFT is needed and the usefulness of PMT can be better highlighted?**
>
> We agree that testing on more challenging tasks. It is important to fully showcase the benefits of PrefixMemoryTuning. The Table 3 and 4 in our paper shows the effectivenss of our method on challenging tasks (e.g., math reasoning and human preference alignment). Table 3 reports a math reasoning setting, in which critique fine-tune [5] Qwen2.5-Math-7B on WebInstruct-CFT with 4K/10K/50K critique pairs and evaluate on AMC’23, AIME’24, and Minerva-Math. Table 4 reports human preference alignment with DPO [7] and SimPO [7]. In both cases, full SFT is substantially stronger than in the low-data regime, yet PrefixMemoryTuning still consistently outperforms SFT and other PEFT baselines, showing that PMT remains beneficial even when the tasks are challenging and SFT is highly competitive.
>
> ---
>
> ***References:***
>
> [1] Xiao, Guangxuan, et al. "Efficient streaming language models with attention sinks." arXiv preprint arXiv:2309.17453 (2023).
>
> [2] Wang, Qifan, et al. "Aprompt: Attention prompt tuning for efficient adaptation of pre-trained language models." Proceedings of the 2023 conference on empirical methods in natural language processing. 2023.
>
> [3] https://github.com/huggingface/peft
>
> [4] Hayou, Soufiane, Nikhil Ghosh, and Bin Yu. "Lora+: Efficient low rank adaptation of large models." arXiv preprint arXiv:2402.12354 (2024).
>
> [5] Wang, Yubo, Xiang Yue, and Wenhu Chen. "Critique fine-tuning: Learning to critique is more effective than learning to imitate." arXiv preprint arXiv:2501.17703 (2025).
>
> [6] Rafailov, Rafael, et al. "Direct preference optimization: Your language model is secretly a reward model." Advances in neural information processing systems 36 (2023): 53728-53741.
>
> [7] Meng, Yu, Mengzhou Xia, and Danqi Chen. "Simpo: Simple preference optimization with a reference-free reward." Advances in Neural Information Processing Systems 37 (2024): 124198-124235.

---

> > ### Comment · Reviewer_UiQ2 · 2025-11-23
> > **Thanks for the response**
> >
> > Thanks for the response and agreeing some of the suggestions as future works. I will keep my rating

---

> > > ### Author Response · Authors · 2025-11-25
> > > **Thank you for your feedback and suggestions**
> > >
> > > Thank you for your time and for your constructive suggestions. We appreciate your feedback!

---

### Author Response · Authors · 2025-12-03
**Summary of Rebuttal Revisions and General Response**

Dear Reviewers and Area Chairs,

Thank you for taking the time to review our paper. We are glad that the reviewers recognize both our analysis of why standard Prefix-Tuning degrades and the value of PrefixMemoryTuning (PMT) for addressing these limitations. After the initial rebuttal, two reviewers (UiQ2, JeiT) ultimately recommended marginally-above-threshold scores (6), and Reviewer gihG stated that they would champion this work, raising their score from 6 to 8. We sincerely appreciate this support! Although Reviewer GSuw remained more critical overall, we carefully addressed their concerns by correcting inaccurate related-work descriptions, expanding it coverage on context-based PEFT and memory views, and clarifying our positioning.

Below we summarize, for each reviewer, the main concerns and what we addressed in the rebuttal and revision:


| Reviewer | Main concerns (very brief) | What we did in rebuttal |
| :--- | :--- | :--- |
| **UiQ2** | Low-data eval may be too easy; no analysis of what "prefix memory" learns; missing attention-sink / Aprompt discussion; wants more challenging tasks. | Clarified the **two-regime design** (low-data in Table 1 vs **rich-data math reasoning & preference alignment** in Tables 3–4), and why SFT can underperform PEFT in low-data while still being strong in richer settings; explained why low-data is important for **IID vs OOD behavior** and isolating the prefix memory effect; discussed the memory view and future work; added **attention-pattern discussion** and a note on attention sinks in Appx. B.2; explained why Aprompt is not included and instead added **LoRA+** as a strong new baseline; emphasized that Tables 3–4 (rich-data regime) already provide challenging setups. |
| **JeiT** | Only small models; missing stronger PEFT baselines (QLoRA / LoRA+); asks if PMT-like transformation could be built into the base architecture. | Explained that the goal is the analysis of PT’s limitations and PMT’s advantages using **widely used, open-source models** (7B/3B) that are feasible on an academic cluster; acknowledged **resource constraints**, noting that evaluating PMT on larger models is future work; added **LoRA+** as a baseline and reported detailed results, showing PMT remains competitive; discussed the idea of **baking the PMT transformation into the base architecture** and clarified that this would require training from scratch, which we mark as a promising but out-of-scope direction. |
| **GSuw** | Inaccurate description of LoRA+; missing references for claim *prefix-tuning excels in low-data*; related work on context-based PEFT / memory incomplete; asks about zero-shot vs few-shot (prompting) vs low-data fine-tuning. | Corrected the **description of LoRA+** in the related-work section to match its actual contribution (differential learning rates for A/B and their theoretical analysis); revised the statement about prefix-tuning in low-data settings and added **explicit citations** showing when prefix-tuning works (Sec. 3.2); **expanded related work** to cover DePT and ADePT as context-based PEFT methods and added a paragraph on the FFN-as-memory perspective; clarified the distinction between our **low-data fine-tuning regime** (Tables 1, 3, 4) and **zero-/few-shot CoT prompting** in “Revisiting Chain-of-Thought Prompting”, and emphasized that **fine-tuning and prompting are orthogonal and potentially complementary**. |
| **gihG** | Only simple feature maps (ELU / GELU) explored; computational cost / scalability for very large LLMs unclear; missing E2VPT-style related work; wants clearer compute & memory overhead vs LoRA; asks about PMT under extremely long contexts. | Added a new **MLP-kernel PMT variant** experiment and reported results on *GoEmotions / DBpedia / BigBench*, showing that MLP kernels are **competitive but not consistently better** (and more expensive), motivating our focus on ELU/GELU as stable feature maps; expanded the **compute analysis (Sec. 6.3)** with training throughput and inference throughput, including **Qwen2.5-72B-Instruct**, where PMT incurs only 0.9% slowdown (7.32 vs 7.39 tok/s); clarified that PMT keeps the attention structure intact and thus remains **compatible with FlashAttention**; added **E2VPT** to the related work discussion; reported **peak fine-tuning memory** comparisons vs LoRA to show similar memory footprint; using the equations in the paper, explained how PMT’s fixed prefix weight + trainable memory design should be more robust than vanilla PT under long input sequences. |

We understand that this year’s ICLR review process is especially demanding, and we are sincerely grateful that you took the time to read our detailed rebuttal and additional experiments. We remain confident that PrefixMemoryTuning (PMT) offers a meaningful contribution to the study of parameter-efficient fine-tuning!

---

### Meta-Review · Area_Chair_YU3n · 2026-01-07

**Summary:**

This submission investigates the performance degradation of Prefix-Tuning (PT) in the context of Large Language Models (LLMs). The authors provide a compelling analysis identifying a fundamental trade-off between the contribution of the parameterized prefix and the input context within the attention mechanism. To address this, they propose **PrefixMemory-Tuning (PMT)**, a novel PEFT architecture that decouples the prefix from the attention head by introducing a trainable memory matrix $M$ that interacts directly with a non-linear mapping of the Query $\phi(q_i)$. Experimental results across several benchmarks, including math reasoning and preference alignment, demonstrate that PMT consistently outperforms vanilla PT and achieves competitive or superior performance compared to LoRA and full fine-tuning.

**Reviewer Concerns:**

The review process saw a significant divergence in opinions. Reviewer gihG emerged as a champion for the work after the rebuttal, praising the scalability and the unified framework. Reviewers UiQ2 and JeiT found the failure analysis of PT convincing and the proposed solution effective, though they noted limitations regarding interpretability and baselines. Reviewer GSuw maintained a negative stance, primarily citing inaccuracies in the related work descriptions and questioning the experimental claims regarding low-data regimes. Through the rebuttal, the authors successfully addressed several technical concerns, leading to an overall positive consensus among the majority of the committee.

**Reviewer Scores:**

The paper presents a solid contribution by revitalizing Prefix-Tuning through a well-justified decoupling strategy. Although the method’s inability to support weight merging is a practical drawback and the baselines could be more modern, the analytical depth regarding PT's failure and the interesting derivation of the memory-based mechanism justify its inclusion in the conference. The authors’ thorough rebuttal and additional experiments effectively addressed the most critical concerns.

---

### Decision · Program_Chairs · 2026-01-26

Accept (Poster)